# Explaining Latent Representations with a Corpus of Examples

**Jonathan Crabbé**
University of Cambridge
jc2133@cam.ac.uk

**Zhaozhi Qian**
University of Cambridge
zq224@maths.cam.ac.uk

**Fergus Imrie**
UCLA
imrie@g.ucla.edu

**Mihaela van der Schaar**
University of Cambridge
The Alan Turing Institute
UCLA
mv472@cam.ac.uk

## Abstract

Modern machine learning models are complicated. Most of them rely on convoluted latent representations of their input to issue a prediction. To achieve greater transparency than a black-box that connects inputs to predictions, it is necessary to gain a deeper understanding of these latent representations. To that aim, we propose SimplEx: a user-centred method that provides example-based explanations with reference to a freely selected set of examples, called the corpus. SimplEx uses the corpus to improve the user's understanding of the latent space with post-hoc explanations answering two questions: (1) Which corpus examples explain the prediction issued for a given test example? (2) What features of these corpus examples are relevant for the model to relate them to the test example? SimplEx provides an answer by reconstructing the test latent representation as a mixture of corpus latent representations. Further, we propose a novel approach, the Integrated Jacobian, that allows SimplEx to make explicit the contribution of each corpus feature in the mixture. Through experiments on tasks ranging from mortality prediction to image classification, we demonstrate that these decompositions are robust and accurate. With illustrative use cases in medicine, we show that SimplEx empowers the user by highlighting relevant patterns in the corpus that explain model representations. Moreover, we demonstrate how the freedom in choosing the corpus allows the user to have personalized explanations in terms of examples that are meaningful for them.

## 1 Introduction and related work

How can we make a machine learning model convincing? If accuracy is undoubtedly necessary, it is rarely sufficient. As these models are used in critical areas such as medicine, finance and the criminal justice system, their black-box nature appears as a major issue [1, 2, 3]. With the necessity to address this problem, the landscape of explainable artificial intelligence (XAI) developed [4, 5]. A first approach in XAI is to focus on *white-box models* that are interpretable by design. However, restricting to a class of inherently interpretable models often comes at the cost of lower prediction accuracy [6]. In this work, we rather focus on *post-hoc explainability* techniques. These methods aim at improving the interpretability of black-box models by complementing their predictions with various kinds of explanations. In this way, it is possible to understand the prediction of a model without sacrificing its prediction accuracy.

35th Conference on Neural Information Processing Systems (NeurIPS 2021).

*Feature importance explanations* are undoubtedly the most widespread type of post-hoc explanations. Popular feature importance methods include SHAP [7, 8, 9], LIME [10], Integrated Gradients [11], Contrastive Examples [12] and Masks [13, 14, 15]. These methods complement the model prediction for an input example with a score attributed to each input feature. This score reflects the importance of each feature for the model to issue its prediction. Knowing which features are important for a model prediction certainly provides more information on the model than the prediction by itself. However, these methods do not provide a reason as to why the model pays attention to these particular features.

Another approach is to contextualize each model prediction with the help of relevant examples. In fact, recent works [16] have demonstrated that human subjects often find example-based explanations more insightful than feature importance explanations. Complementing the model's predictions with relevant examples previously seen by the model is commonly known as *Case-Based Reasoning* (CBR) [17, 18, 19]. The implementations of CBR generally involve models that create a synthetic representation of the dataset, where examples with similar patterns are summarized by prototypes [20, 21, 22]. At inference time, these models relate new examples to one or several prototypes to issue a prediction. In this way, the patterns that are used by the model to issue a prediction are made explicit with the help of relevant prototypes. A limitation of this approach is the restricted model architecture. The aforementioned procedure requires to opt for a family of models that rely on prototypes to issue a prediction. This family of model might not always be the most suitable for the task at hand. This motivates the development of generic post-hoc methods that make few or no assumption on the model.

The most common approach to provide example-based explanations for a wide variety of models mirrors feature importance methods. The idea is to complement the model prediction by attributing a score to each training example. This score reflects the importance of each training example for the model to issue its prediction. This score will typically be computed by simulating the effect of removing each training instance from the training set on the learned model [23]. Popular examples of such methods include Influence Functions [24] and Data-Shapley [25, 26]. These methods offer the advantage of being flexible enough to be used with a wide variety of models. They produce scores that describe what the model could have predicted if some examples were absent from the training set. This is very interesting in a data valuation perspective. However, in an explanation perspective, it is not clear how to reconstruct the model predictions with these importance scores.

So far, we have only discussed works that provide explanations of a model output, which is the tip of the iceberg. Modern machine learning models involve many convoluted transformations to deduce the output from an input. These transformations are expressed in terms of intermediate variables that are often called *latent variables*. Some treatment of these latent variables is necessary if we want to provide explanations that take the model complexity into account. This motivates several works that push the explainability task beyond the realm of model outputs. Among the most noticeable contributions in this endeavour, we cite *Concept Activation Vectors* that create a dictionary between human friendly concepts (such as the presence of stripes in an image) and their representation in terms of latent vectors [27]. Another interesting contribution is the *Deep k-Nearest Neighbors* model that contextualizes the prediction for an example with its Nearest Neighbours in the space of latent variables, the *latent space* [28]. An alternative exploration of the latent space is offered by the *representer theorem* that allows, under restrictive assumptions, to use latent vectors to decompose a model's prediction in terms of its training examples [29].

**Contribution** In this work, we introduce a novel approach called SimplEx that lies at the crossroad of the above research directions. SimplEx outputs post-hoc explanations in the form of Figure 1, where the model's prediction and latent representation for a test example is approximated as a mixture of examples extracted from a *corpus* of examples.

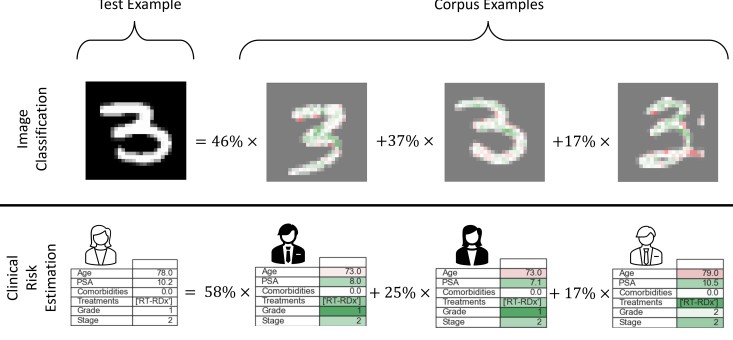

Figure 1: An example of corpus decomposition with SimplEx.

In each case, SimplEx highlights the role played by each feature of each corpus example in the latent space decomposition. SimplEx centralizes many functionalities that, to the best of our knowledge, constitute a leap forward from the previous state of the art. (1) SimplEx gives the user freedom to choose the corpus of examples whom with the model's predictions are decomposed. Unlike previous methods such as the representer theorem, there is no need for this corpus of examples to be equal to the model's training set. This is particularly interesting for two reasons: (a) the training set of a model is not always accessible (b) the user might want explanations in terms of examples that make sense for them. For instance, a doctor might want to understand the predictions of a risk model in terms of patients they know. (2) The decompositions of SimplEx are valid, both in latent and output space. We show that, in both cases, the corpus mixtures discovered by SimplEx offer significantly more precision and robustness than previous methods such as Deep k-Nearest Neighbors and the representer theorem. (3) SimplEx details the role played by each feature in the corpus mixture. This is done by introducing Integrated Jacobians, a generalization of Integrated Gradients that makes the contribution of each corpus feature explicit in the latent space decomposition. This creates a bridge between two research directions that have mostly developed independently: feature importance and example-based explanations [19, 30]. In Section 3 of the supplementary material, we report a user-study involving 10 clinicians. This study supports the significance of our contribution.

## 2 SimplEx

In this section, we formulate our method rigorously. Our purpose is to explain the black-box prediction for an unseen test example with the help of a set of known examples that we call the *corpus*. We start with a clear statement of the family of black-boxes for which our method applies. Then, we detail how the set of corpus examples can be used to decompose a black-box representation for the unseen example. Finally, we show that the corpus decomposition can offer explanations at the feature level.

### 2.1 Preliminaries

Let $\mathcal{X} \subseteq \mathbb{R}^{d_X}$ be an input (or feature) space and $\mathcal{Y} \subseteq \mathbb{R}^{d_Y}$ be an output (or label) space, where $d_X$ and $d_Y$ are respectively the dimension of the input and the output space. Our task is to explain individual predictions of a given black-box $\mathbf{f} : \mathcal{X} \rightarrow \mathcal{Y}$. In order to build our explainability method, we need to make an assumption on the family of black-boxes that we wish to interpret.

**Assumption 2.1** (Black-box Restriction). We restrict to black-boxes $\mathbf{f} : \mathcal{X} \rightarrow \mathcal{Y}$ that can be decomposed as $\mathbf{f} = \mathbf{l} \circ \mathbf{g}$, where $\mathbf{g} : \mathcal{X} \rightarrow \mathcal{H}$ maps an input $\mathbf{x} \in \mathcal{X}$ to a latent vector $\mathbf{h} = \mathbf{g}(\mathbf{x}) \in \mathcal{H}$ and $\mathbf{l} : \mathcal{H} \rightarrow \mathcal{Y}$ linearly maps[1] a latent vector $\mathbf{h} \in \mathcal{H}$ to an output $\mathbf{y} = \mathbf{l}(\mathbf{h}) = \mathbf{A}\mathbf{h} \in \mathcal{Y}$. In the following, we call $\mathcal{H} \subseteq \mathbb{R}^{d_H}$ the *latent space*. Typically, this space has higher dimension than the output space $d_H > d_Y$.

*Remark* 2.1. In the context of deep-learning, this assumption requires that the last hidden layer maps linearly to the output. While it is often the case, it is crucial in the following since we will use the fact that linear combinations in latent space correspond to linear combinations in output space. Our purpose is to gain insights on the structure of the latent space.

*Remark* 2.2. This assumption is compatible with regression and classification models, we just need to clarify what we mean by *output* in the case of classification. If $\mathbf{f}$ is a classification black-box that predicts the probabilities for each class, it will typically take the form in Assumption 2.1 up to a normalizing map $\phi$ (typically a softmax): $\mathbf{f} = \phi \circ \mathbf{l} \circ \mathbf{g}$. In this case, we ignore[2] the normalizing map $\phi$ and define the output to be $\mathbf{y} = (\mathbf{l} \circ \mathbf{g})(\mathbf{x})$.

Our explanations for $\mathbf{f}$ rely on a set of examples that we call the corpus. These examples will typically (but not necessarily) be a representative subset of the black-box training set. The corpus set has to be understood as a set of reference examples that we want to use as building blocks to interpret unseen examples. In order to index these examples, it will be useful to denote by $[n_1 : n_2]$ the set of natural numbers between the natural numbers $n_1$ and $n_2$ with $n_1 < n_2$. Further, we denote $[n] = [1 : n]$ the set of natural numbers between 1 and $n \geq 1$. The corpus of examples is a set $\mathcal{C} = \{\mathbf{x}^c \mid c \in [C]\}$ containing $C \in \mathbb{N}^*$ examples $\mathbf{x}^c \in \mathcal{X}$. In the following, superscripts are labels for examples and subscripts are labels for vector components. In this way, $x_i^c$ has to be understood as the component $i$ of corpus example $c$.

---

[1] The map can in fact be affine. In the following, we omit the bias term $\mathbf{b} \in \mathcal{Y}$ that can be reabsorbed in $\mathbf{g}$.
[2] There is no loss of information as the output allows us to reconstruct class probabilities $\mathbf{p}$ via $\mathbf{p} = \phi(\mathbf{y})$.

## 2.2 A corpus of examples to explain a latent representation

Our purpose is to understand a prediction $\mathbf{f}(\mathbf{x})$ for an unseen test example $\mathbf{x}$ with the help of the corpus. How can we decompose the prediction $\mathbf{f}(\mathbf{x})$ in terms of corpus predictions $\mathbf{f}(\mathbf{x}^c)$? A naive attempt would be to express $\mathbf{x}$ as a mixture of inputs from the corpus $\mathcal{C}$: $\mathbf{x} = \sum_{c=1}^{C} w^c \mathbf{x}^c$ with weights $w^c \in [0,1]$ that sum to one $\sum_{c=1}^{C} w^c = 1$. The weakness of this approach is that the signification of the mixture weights is not conserved if the black-box $\mathbf{f}$ is not a linear map: $\mathbf{f}(\sum_{c=1}^{C} w^c \mathbf{x}^c) \neq \sum_{c=1}^{C} w^c \mathbf{f}(\mathbf{x}^c)$.

Fortunately, Assumption 2.1 offers us a better vector space to perform a corpus decomposition of the unseen example $\mathbf{x}$. We first note that the map $\mathbf{g}$ induces a latent representation of the corpus $\mathbf{g}(\mathcal{C}) = \{\mathbf{h}^c = \mathbf{g}(\mathbf{x}^c) \mid \mathbf{x}^c \in \mathcal{C}\} \subset \mathcal{H}$. Similarly, $\mathbf{x}$ has a latent representation $\mathbf{h} = \mathbf{g}(\mathbf{x}) \in \mathcal{H}$. Following the above line of reasoning, we could therefore perform a corpus decomposition in latent space $\mathbf{h} = \sum_{c=1}^{C} w^c \mathbf{h}^c$. Now, by using the linearity of $\mathbf{l}$, we can compute the black-box output of this mixture in latent space: $\mathbf{l}(\sum_{c=1}^{C} w^c \mathbf{h}^c) = \sum_{c=1}^{C} w^c \mathbf{l}(\mathbf{h}^c)$. In this case, the weights that are used to decompose the latent representation $\mathbf{h}$ in terms of the latent representation of the corpus $\mathbf{g}(\mathcal{C})$ also reflect the way in which the black-box prediction $\mathbf{f}(\mathbf{x})$ can be decomposed in terms of the corpus outputs $\mathbf{f}(\mathcal{C})$. This hints that the latent space $\mathcal{H}$ is endowed with the appropriate geometry to make corpus decompositions. More formally, we think in terms of the convex hull spanned by the corpus.

**Definition 2.1** (Corpus Hull). The *corpus convex hull* spanned by a corpus $\mathcal{C}$ with latent representation $\mathbf{g}(\mathcal{C}) = \{\mathbf{h}^c = \mathbf{g}(\mathbf{x}^c) \mid \mathbf{x}^c \in \mathcal{C}\} \subset \mathcal{H}$ is the convex set

$$\mathcal{CH}(\mathcal{C}) = \left\{ \sum_{c=1}^{C} w^c \mathbf{h}^c \ \middle| \ w^c \in [0,1] \ \forall c \in [C] \ \wedge \ \sum_{c=1}^{C} w^c = 1 \right\}.$$

*Remark* 2.3. This is the set of latent vectors that are a mixture of the corpus latent vectors.

At this stage, it is important to notice that an exact corpus decomposition is not possible if $\mathbf{h} \notin \mathcal{CH}(\mathcal{C})$. In such a case, the best we can do is to find the element $\hat{\mathbf{h}} \in \mathcal{CH}(\mathcal{C})$ that best approximates $\mathbf{h}$. If $\mathcal{H}$ is endowed with a norm $\|\cdot\|_{\mathcal{H}}$, this corresponds to the convex optimization problem

$$\hat{\mathbf{h}} = \underset{\tilde{\mathbf{h}} \in \mathcal{CH}(\mathcal{C})}{\arg\min} \ \| \mathbf{h} - \tilde{\mathbf{h}} \|_{\mathcal{H}} . \tag{1}$$

By definition, the corpus representation $\hat{\mathbf{h}}$ of $\mathbf{h}$ can be expanded[3] as a mixture of elements from $\mathbf{g}(\mathcal{C})$: $\hat{\mathbf{h}} = \sum_{c=1}^{C} w^c \mathbf{h}^c$. The weight can naturally be interpreted as a measure of importance in the reconstruction of $\mathbf{h}$ with the corpus. Clearly, $w^c \approx 0$ for some $c \in [C]$ indicates that $\mathbf{h}^c$ does not play a significant role in the corpus representation $\hat{\mathbf{h}}$ of $\mathbf{h}$. On the other hand, $w^c \approx 1$ indicates that $\mathbf{h}^c$ generates the corpus representation $\hat{\mathbf{h}}$ by itself.

At this stage, a natural question arises: how can we know if the corpus approximation $\hat{\mathbf{h}}$ is a good approximation for $\mathbf{h}$? The answer is given by the residual vector $\mathbf{h} - \hat{\mathbf{h}}$ that measures the shift between the latent representation $\mathbf{h} = \mathbf{g}(\mathbf{x})$ and the corpus hull $\mathcal{CH}(\mathcal{C})$. It is natural to use this residual vector to detect examples that cannot be explained with the selected corpus of examples $\mathcal{C}$.

**Definition 2.2** (Corpus Residual). The *corpus residual* associated to a latent vector $\mathbf{h} \in \mathcal{H}$ and its corpus representation $\hat{\mathbf{h}} \in \mathcal{CH}(\mathcal{C})$ solving (1) is the quantity

$$r_{\mathcal{C}}(\mathbf{h}) = \| \mathbf{h} - \hat{\mathbf{h}} \|_{\mathcal{H}} = \underset{\tilde{\mathbf{h}} \in \mathcal{CH}(\mathcal{C})}{\min} \ \| \mathbf{h} - \tilde{\mathbf{h}} \|_{\mathcal{H}} .$$

In Section 1.1 of the supplementary material, we show that the corpus residual also controls the quality of the corpus approximation in output space $\mathcal{Y}$. All the corpus-related quantities that we have introduced so far are summarized visually in Figure 2. Note that this Figure is a simplification of the reality as $C$ will typically

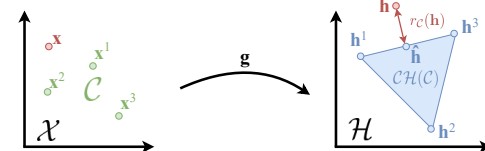

Figure 2: Corpus convex hull and residual.

be larger than 3 and $d_X, d_H$ will typically be higher than 2. We are now endowed with a rigorous way to decompose a test example in terms of corpus examples *in latent space*. In the next section, we detail how to pull-back this decomposition to input space.

---

[3] Note that this decomposition might not be unique, more details in Section 1 of the supplementary material.

## 2.3 Transferring the corpus explanation in input space

Now that we are endowed with a corpus decomposition $\hat{\mathbf{h}} = \sum_{c=1}^{C} w^c \mathbf{h}^c$ that approximates $\mathbf{h}$, it would be convenient to have an understanding of the corpus decomposition in input space $\mathcal{X}$. For the sake of notation, we will assume that the corpus approximation is good so that it is unnecessary to draw a distinction between the latent representation $\mathbf{h}$ of the unseen example $\mathbf{x}$ and its corpus decomposition $\hat{\mathbf{h}}$. If we want to understand the corpus decomposition in input space, a natural approach [11] is to fix a baseline input $\mathbf{x}^0$ together with its latent representation $\mathbf{h}^0 = \mathbf{g}(\mathbf{x}^0)$. Let us now decompose the representation shift $\mathbf{h} - \mathbf{h}^0$ in terms of the corpus:

$$\mathbf{h} - \mathbf{h}^0 = \sum_{c=1}^{C} w^c \left( \mathbf{h}^c - \mathbf{h}^0 \right). \tag{2}$$

With this decomposition, we understand the total shift in latent space $\mathbf{h} - \mathbf{h}^0$ in terms of individual contributions from each corpus member. In the following, we focus on the comparison between the baseline and a single corpus example $\mathbf{x}^c$ together with its latent representation $\mathbf{h}^c$ by keeping in mind that the full decomposition (2) can be reconstructed with the whole corpus. To bring the discussion in input space $\mathcal{X}$, we interpret the shift in latent space $\mathbf{h}^c - \mathbf{h}^0$ as resulting from a shift $\mathbf{x}^c - \mathbf{x}^0$ in the input space. We are interested in the contribution of each feature to the latent space shift. To decompose the shift in latent space in terms of the features, we parametrize the shift in input space with a line $\boldsymbol{\gamma}^c : [0,1] \to \mathcal{X}$ that goes from the baseline to the corpus example: $\boldsymbol{\gamma}^c(t) = \mathbf{x}^0 + t \cdot (\mathbf{x}^c - \mathbf{x}^0)$ for $t \in [0,1]$. Together with the black-box, this line induces a curve in latent space $\mathbf{g} \circ \boldsymbol{\gamma}^c : [0,1] \to \mathcal{H}$ that goes from the baseline latent representation $\mathbf{h}^0$ to the corpus example latent representation $\mathbf{h}^c$. Let us now use an infinitesimal decomposition of this curve to make the contribution of each input feature explicit. If we assume that $\mathbf{g}$ is differentiable at $\boldsymbol{\gamma}^c(t)$, we can use a first order approximation of the curve at the vicinity of $t \in (0,1)$ to decompose the infinitesimal shift in latent space:

$$\underbrace{\mathbf{g} \circ \boldsymbol{\gamma}^c(t + \delta t) - \mathbf{g} \circ \boldsymbol{\gamma}^c(t)}_{\text{Infinitesimal shift in latent space}} = \sum_{i=1}^{d_X} \left. \frac{\partial \mathbf{g}}{\partial x_i} \right|_{\boldsymbol{\gamma}^c(t)} \left. \frac{d\gamma_i^c}{dt} \right|_t \delta t + o(\delta t)$$

$$= \sum_{i=1}^{d_X} \left. \frac{\partial \mathbf{g}}{\partial x_i} \right|_{\boldsymbol{\gamma}^c(t)} (x_i^c - x_i^0) \cdot \delta t + o(\delta t),$$

where we used $\gamma_i^c(t) = x_i^0 + t \cdot (x_i^c - x_i^0)$ to obtain the second equality. In this decomposition, each input feature contributes additively to the infinitesimal shift in latent space. It follows trivially that the contribution of the input feature corresponding to input dimension $i \in [d_X]$ is given by

$$\delta \mathbf{j}_i^c(t) = (x_i^c - x_i^0) \cdot \left. \frac{\partial \mathbf{g}}{\partial x_i} \right|_{\boldsymbol{\gamma}^c(t)} \delta t \qquad \in \mathcal{H}.$$

In order to compute the overall contribution of feature $i$ to the shift, we let $\delta t \to 0$ and we sum the infinitesimal contributions along the line $\boldsymbol{\gamma}^c$. If we assume[4] that $\mathbf{g}$ is almost everywhere differentiable, this sum converges to an integral in the limit $\delta t \to 0$. This motivates the following definitions.

**Definition 2.3** (Integrated Jacobian & Projection). The *integrated Jacobian* between a baseline $(\mathbf{x}^0, \mathbf{h}^0 = \mathbf{g}(\mathbf{x}^0))$ and a corpus example $(\mathbf{x}^c, \mathbf{h}^c = \mathbf{g}(\mathbf{x}^c)) \in \mathcal{X} \times \mathcal{H}$ associated to feature $i \in [d_X]$ is

$$\mathbf{j}_i^c = \left( x_i^c - x_i^0 \right) \int_0^1 \left. \frac{\partial \mathbf{g}}{\partial x_i} \right|_{\boldsymbol{\gamma}^c(t)} dt \qquad \in \mathcal{H},$$

where $\boldsymbol{\gamma}^c(t) \equiv \mathbf{x}^0 + t \cdot \left( \mathbf{x}^c - \mathbf{x}^0 \right)$ for $t \in [0,1]$. This vector indicates the shift in latent space induced by feature $i$ of corpus example $c$ when comparing the corpus example with the baseline. To summarize this contribution to the shift $\mathbf{h} - \mathbf{h}^0$ described in (2), we define the *projected Jacobian*

$$p_i^c = \text{proj}_{\mathbf{h} - \mathbf{h}^0} \left( \mathbf{j}_i^c \right) \equiv \frac{\langle \, \mathbf{h} - \mathbf{h}^0 \, , \, \mathbf{j}_i^c \, \rangle}{\langle \, \mathbf{h} - \mathbf{h}^0 \, , \, \mathbf{h} - \mathbf{h}^0 \, \rangle} \qquad \in \mathbb{R},$$

where $\langle \cdot, \cdot \rangle$ is an inner product for $\mathcal{H}$ and the normalization is chosen for the purpose of Proposition 2.1.

*Remark* 2.4. The integrated Jacobian can be seen as a latent-space generalization of Integrated Gradients [11]. In Section 1.3 of the supplementary material, we establish the relationship between the two quantities: $\text{IG}_i^c = \text{l}(\mathbf{j}_i^c)$.

---

[4]This is not restrictive, DNNs with ReLU activation functions satisfy this assumption for instance.

We summarize the Jacobian quantities in Figure 3. By inspecting the figure, we notice that projected Jacobians encode the contribution of feature $i$ from corpus example $c$ to the overall shift in latent space: $p_i^c > 0$ implies that this feature creates a shift pointing in the same direction as the overall shift; $p_i^c < 0$ implies that this feature creates a shift pointing in the opposite direction and $p_i^c = 0$ implies that this feature creates a shift in an orthogonal direction. We use the projections to summarize the contribution of each feature in Figures 1, 8 & 9. The colors green and red indicate respectively a positive and negative projection. In addition to these geometrical insights, Jacobian quantities come with natural properties.

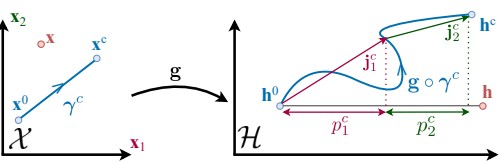

Figure 3: Integrated Jacobian and projection.

**Proposition 2.1** (Properties of Integrated Jacobians). *Consider a baseline $(\boldsymbol{x}^0, \boldsymbol{h}^0 = \boldsymbol{g}(\boldsymbol{x}^0))$ and a test example together with their latent representation $(\boldsymbol{x}, \boldsymbol{h} = \boldsymbol{g}(\boldsymbol{x})) \in \mathcal{X} \times \mathcal{H}$. If the shift $\boldsymbol{h} - \boldsymbol{h}^0$ admits a decomposition (2), the following properties hold.*

$$(A): \sum_{c=1}^{C}\sum_{i=1}^{d_X} w^c \boldsymbol{j}_i^c = \boldsymbol{h} - \boldsymbol{h}^0 \qquad (B): \sum_{c=1}^{C}\sum_{i=1}^{d_X} w^c p_i^c = 1.$$

*Proof.* The proof is provided in Section 1.4 of the supplementary material. □

These properties show that the integrated Jacobians and their projections are the quantities that we are looking for: they transfer the corpus explanation into input space. The first equality decomposes the shift in latent space in terms of contributions $w^c \boldsymbol{j}_i^c$ arising from each feature of each corpus example. The second equality sets a natural scale to the contribution of each feature. For this reason, it is natural to use $w^c p_i^c$ to measure the contribution of feature $i$ of corpus example $c$.

## 3 Experiments

In this section, we evaluate quantitatively several aspects of our method. In a first experiment, we verify that the corpus decomposition scheme described in Section 2 yields good approximations for the latent representation of test examples extracted from the same dataset as the corpus examples. In a realistic clinical use case, we illustrate the usage of SimplEx in a set-up where different corpora reflecting different datasets are used. The experiments are summarized below. In Section 2 of the supplementary material, we provide more details and further experiments with time series and synthetic data. The code for our method and experiments is available on the Github repository https://github.com/JonathanCrabbe/Simplex. All the experiments have been replicated on different machines.

### 3.1 Precision of corpus decomposition

**Description** The purpose of this experiment is to check if the corpus decompositions described in Section 2 allows us to build good approximations of the latent representation of test examples. We start with a dataset $\mathcal{D}$ that we split into a training set $\mathcal{D}_{\text{train}}$ and a testing set $\mathcal{D}_{\text{test}}$. We train a black-box $\mathbf{f}$ for a given task on the training set $\mathcal{D}_{\text{train}}$. We randomly sample a set of corpus examples from the training set $\mathcal{C} \subset \mathcal{D}_{\text{train}}$ (we omit the true labels for the corpus examples) and a set of test examples from the testing set $\mathcal{T} \subset \mathcal{D}_{\text{test}}$. For each test example $\mathbf{x} \in \mathcal{T}$, we build an approximation $\hat{\mathbf{h}}$ for $\mathbf{h} = \mathbf{g}(\mathbf{x})$ with the corpus examples latent representations. In each case, we let the method use only $K$ corpus examples to build the approximation. We repeat the experiment for several values of $K$.

**Metrics** We are interested in measuring the precision of the corpus approximation in latent space and in output space. To that aim, we use the $R^2$ score in both spaces. In this way, $R_{\mathcal{H}}^2$ measures the precision of the corpus approximation $\hat{\mathbf{h}}$ with respect to the true latent representation $\mathbf{h}$. Similarly, $R_{\mathcal{Y}}^2$ measures the precision of the corpus approximation $\hat{\mathbf{y}} = \mathbf{l}(\hat{\mathbf{h}})$ with respect to the true output $\mathbf{y} = \mathbf{l}(\mathbf{h})$. Both of these metrics satisfy $-\infty < R^2 \leq 1$. A higher $R^2$ score is better with $R^2 = 1$

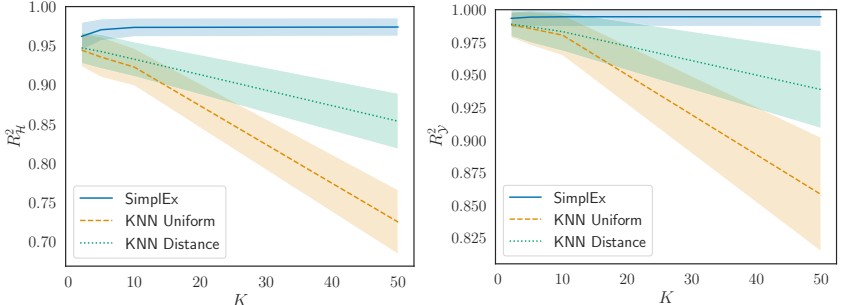

(a) $R^2_{\mathcal{H}}$ score for the latent approximation  (b) $R^2_{\mathcal{Y}}$ score for the output approximation

Figure 4: Precision of corpus decomposition for prostate cancer (avg $\pm$ std).

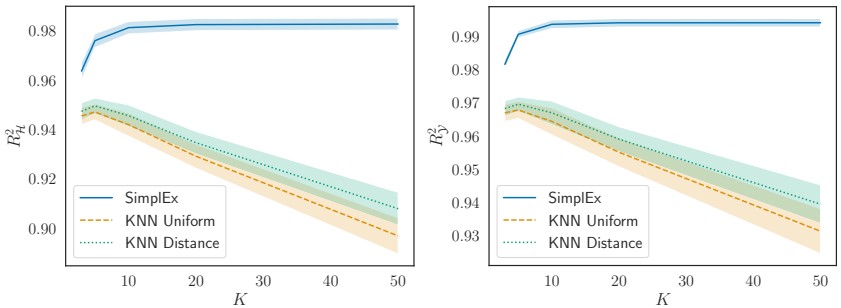

(a) $R^2_{\mathcal{H}}$ score for the latent approximation  (b) $R^2_{\mathcal{Y}}$ score for the output approximation

Figure 5: Precision of corpus decomposition for MNIST (avg $\pm$ std).

corresponding to a perfect approximation. All the metrics are computed over the test examples $\mathcal{T}$. The experiments are repeated 10 times to report standard deviations across different runs.

**Baselines** We compare our method[5] (SimplEx) with 3 baselines. A first approach, inspired by [28], consists in using the $K$-nearest corpus neighbours *in latent space* to build the latent approximation $\hat{\mathbf{h}}$. Building on this idea, we introduce two baselines (1) KNN Uniform that takes the average latent representation of the $K$-nearest corpus neighbours of $\mathbf{h}$ in latent space (2) KNN Distance that computes the same average with weights $w^c$ inversely proportional to the distance $\| \mathbf{h} - \mathbf{h}^c \|_{\mathcal{H}}$. Finally, we use the representer theorem [29] to produce an approximation $\hat{\mathbf{y}}$ of $\mathbf{y}$ with the corpus $\mathcal{C}$. Unlike the other methods, the representer theorem does not allow to produce an approximation in latent space.

**Datasets** We use two different datasets with distinct tasks for our experiment: (1) 240,486 patients enrolled in the American SEER program [31]. We consider the binary classification task of predicting cancer mortality for patients with prostate cancer. We train a multilayer perceptron (MLP) for this task. Since this task is simple, we show that a corpus of $C = 100$ patients yields good approximations. (2) 70,000 MNIST images of handwritten digits [32]. We consider the multiclass classification task of identifying the digit represented on each image. We train a convolutional neural network (CNN) for the image classification. This classification task is more complex than the previous one (higher $d_X$ and $d_Y$), we show that a corpus of $C = 1,000$ images yields good approximations in this case.

**Results** The results for SimplEx and the KNN baselines are presented in Figure 4 & 5. Several things can be deduced from these results: (1) It is generally harder to produce a good approximation in latent space than in output space as $R^2_{\mathcal{H}} < R^2_{\mathcal{Y}}$ for most examples (2) SimplEx produces the most accurate approximations, both in latent and output space. These approximations are of high quality with $R^2 \approx 1$. (3) The trends are qualitatively different between SimplEx and the other baselines. The accuracy of SimplEx increases with $K$ and stabilizes when a small number of corpus members contribute ($K = 5$ in both cases). The accuracy of the KNN baselines increases with $K$,

---

[5]To enforce SimplEx to select $K$ examples, we add a $L^1$ penalty making the $C - K$ smallest weights vanish.

reaches a maximum for a small $K$ and steadily decreases for larger $K$. This can be understood easily: when $K$ increases beyond the number of relevant corpus examples, irrelevant examples will be added in the decomposition. SimplEx will typically annihilate the effect of these irrelevant examples by setting their weights $w^c$ to zero in the corpus decomposition. The KNN baselines include the irrelevant corpus members in the decomposition, which alters the quality of the approximation. This suggests that $K$ has to be tuned for each example with KNN baselines, while the optimal number of corpus examples to contribute is learned by SimplEx. (4) The standard deviations indicate that the performances of SimplEx are more consistent across different runs. This is particularly true in the prostate cancer experiment, where the corpus size $C$ is smaller. This suggests that SimplEx is more robust than the baselines. (5) For the representer theorem, we have $R_{\mathcal{Y}}^2 = -(6.6 \pm 6.1) \cdot 10^7$ for the prostate cancer dataset and $R_{\mathcal{Y}}^2 = -(7.2 \pm 6.6)$ for MNIST. This corresponds to poor estimations of the black-box output. We propose some hypotheses to explain this observation in Section 2.1 of the supplementary material.

### 3.2 Significance of Jacobian Projections

**Description** The purpose of this experiment is to check if SimplEx's Jacobian Projections are a good measure of the importance for each corpus feature in constructing the test latent representation $\mathbf{h}$. In the same setting as in the previous experiment, we start with a corpus $\mathcal{C}$ of $C = 500$ MNIST images. We build a corpus approximation for an example $\mathbf{x} \in \mathcal{X}$ with latent representation $\mathbf{h} = \mathbf{g}(\mathbf{x}) \in \mathcal{H}$. The precision of this approximation is reflected by its corpus residual $r_{\mathcal{C}}(\mathbf{h})$. For each corpus example $\mathbf{x}^c \in \mathcal{C}$, we would like to identify the features that are the most important in constructing the corpus decomposition of $\mathbf{h}$. With SimplEx, this is reflected by the Jacobian Projections $p_i^c$. We evaluate these scores for each feature $i \in [d_X]$ of each corpus example $c \in [C]$. For each corpus image $\mathbf{x}^c \in \mathcal{C}$, we select the $n$ most important pixels according to the Jacobian Projections and the baseline. In each case, we build a mask $\mathbf{m}^c$ that replaces these $n$ most important pixels by black pixels. This yields a corrupted corpus image $\mathbf{x}_{cor}^c = \mathbf{m}^c \odot \mathbf{x}^c$, where $\odot$ denotes the Hadamard product. By corrupting all the corpus images, we obtain a corrupted corpus $\mathcal{C}_{cor}$. We analyse how well this corrupted corpus approximates $\mathbf{h}$, this yields a residual $r_{\mathcal{C}_{cor}}(\mathbf{h})$.

**Metric** We are interested in measuring the effectiveness of the corpus corruption. This is reflected by the metric $\delta_{cor}(\mathbf{h}) = r_{\mathcal{C}_{cor}}(\mathbf{h}) - r_{\mathcal{C}}(\mathbf{h})$. A higher value for this metric indicates that the features selected by the saliency method are more important for the corpus to produce a good approximation of $\mathbf{h}$ in latent space. We repeat this experiment for 100 test examples and for different numbers $n$ of perturbed pixels.

**Baseline** As a baseline for our experiment, we use Integrated Gradients, which is close in spirit to our method. In a similar fashion, we compute the Integrated Gradients $IG_i^c$ for each feature $i \in [d_X]$ of each corpus example $c \in [C]$ and construct a corrupted corpus based on these scores.

**Results** The results are presented in the form of box plots in Figure 6. We observe that the corruptions induced by the Jacobian Projections are significantly more impactful when few pixels are perturbed. The two methods become equivalent when more pixels are perturbed. This demonstrates that Jacobian Projections are more suitable to measure the importance of features when performing a latent space reconstruction, as it is the case for SimplEx.

### 3.3 Use case: clinical risk model across countries

Very often, clinical risk models are produced and validated with the data of patients treated at a single site [33]. This can cause problems when these models are deployed at different sites for two reasons: (1) Patients from different sites can have different characteristics (2) Rules that are learned for one site might not be true for another site. One possible way to alleviate this problem would be to detect patients for which the model prediction is highly extrapolated and/or ambiguous. In this way, doctors from different sites can make an enlightened use of the risk model rather than blindly believing the model's predictions. We demonstrate that SimplEx provides a natural framework for this set-up.

As in the previous experiment, we consider a dataset $\mathcal{D}_{\text{USA}}$ containing patients enrolled in the American SEER program [31]. We train and validate an MLP risk model with $\mathcal{D}_{\text{USA}}$. To give a realistic realization of the above use-case, we assume that we want to deploy this risk model in a different site: the United Kingdom. For this purpose, we extract $\mathcal{D}_{\text{UK}}$ from the set of 10,086 patients enrolled in the British Prostate Cancer UK program [34]. These patients are characterized by the

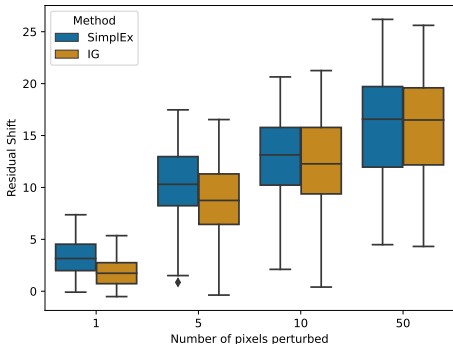

Figure 6: Increase in the corpus residual caused by each method (higher is better).

same features for both $\mathcal{D}_{\text{UK}}$ and $\mathcal{D}_{\text{USA}}$. However, the datasets $\mathcal{D}_{\text{UK}}$ and $\mathcal{D}_{\text{USA}}$ differ by a covariate shift: patients from $\mathcal{D}_{\text{UK}}$ are in general older and at earlier clinical stages.

When comparing the two populations in terms of the model, a first interesting question to ask is whether the covariate shift between $\mathcal{D}_{\text{USA}}$ and $\mathcal{D}_{\text{UK}}$ affects the model representation. To explore this question, we take a first corpus of American patients $\mathcal{C}_{\text{USA}} \subset \mathcal{D}_{\text{USA}}$. If there is indeed a difference in terms of the latent representations, we expect the representations of test examples from $\mathcal{D}_{\text{UK}}$ to be less closely approximated by their decomposition with respect to $\mathcal{C}_{\text{USA}}$. If this is true, the corpus residuals associated to examples of $\mathcal{D}_{\text{UK}}$ will typically larger than the ones associated to $\mathcal{D}_{\text{USA}}$. To evaluate this quantitatively, we consider a mixed set of test examples $\mathcal{T}$ sampled from both $\mathcal{D}_{\text{UK}}$ and $\mathcal{D}_{\text{USA}}$: $\mathcal{T} \subset \mathcal{D}_{\text{UK}} \sqcup \mathcal{D}_{\text{USA}}$. We sample 100 examples from both sources: $|\mathcal{T} \cap \mathcal{D}_{\text{UK}}| = |\mathcal{T} \cap \mathcal{D}_{\text{USA}}| = 100$. We then approximate the latent representation of each example $\mathbf{h} \in \mathbf{g}(\mathcal{T})$ and compute the associated corpus residual $r_{\mathcal{C}_{\text{USA}}}(\mathbf{h})$. We sort the test examples from $\mathcal{T}$ by decreasing order of corpus residual and we use this sorted list to see if we can detect the examples from $\mathcal{D}_{\text{UK}}$. We use previous baselines for comparison, results are shown in Figure 7.

Several things can be deduced from this experiment. (1) The results strongly suggest that the difference between the two datasets $\mathcal{D}_{\text{USA}}$ and $\mathcal{D}_{\text{UK}}$ is reflected in their latent representations. (2) The corpus residuals from SimplEx offer the most reliable way to detect examples that are different from the corpus examples $\mathcal{C}_{\text{USA}}$. None of the methods matches the maximal baseline since some examples of $\mathcal{D}_{\text{USA}}$ resemble examples from $\mathcal{D}_{\text{UK}}$. (3) When the corpus examples are representative of the training set, as it is the case in the experiment, our approach based on SimplEx provides a systematic way to detect test examples that have representations that are different from the ones produced at training time. A doctor should be more sceptical with respect to model predictions associated to larger residual with respect to $\mathcal{C}_{\text{USA}}$ as these arise from an extrapolation region of the latent space.

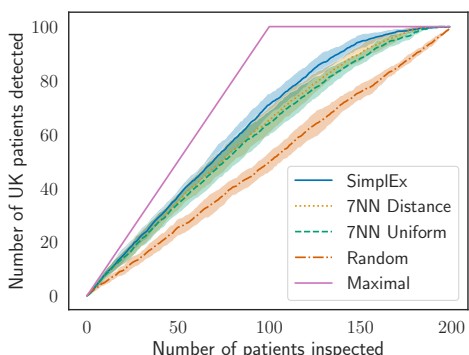

Figure 7: Detecting UK patients (avg.±std.).

Let us now make the case more concrete. Suppose that an American and a British doctor use the above risk model to predict the outcome for their patients. Each doctor wants to decompose the predictions of the model in terms of patients they know. Hence, the American doctor selects a corpus of American patients $\mathcal{C}_{\text{USA}} \subset \mathcal{D}_{\text{USA}}$ and the British doctor selects a corpus of British patients $\mathcal{C}_{\text{UK}} \subset \mathcal{D}_{\text{UK}}$. Both corpora have the same size $C_{\text{USA}} = C_{\text{UK}} = 1,000$. We suppose that the doctors know the model prediction and the true outcome for each patient in their corpus. Both doctors are sceptical about the risk model and want to use SimplEx to decide when it can be trusted. This leads them to a natural question: is it possible to anticipate misclassification with the help of SimplEx?

In Figure 8 & 9, we provide two typical examples of misclassified British patients from $\mathcal{D}_{\text{UK}} \setminus \mathcal{C}_{\text{UK}}$ together with their decomposition in terms of the two corpora $\mathcal{C}_{\text{USA}}$ and $\mathcal{C}_{\text{UK}}$. These two examples

exhibit two qualitatively different situations. In Figure 8, both the American and the British doctors make the same observation: the model relates the test patient to corpus patients that are mostly misclassified by the model. With the help of SimplEx, both doctors will rightfully be sceptical with respect to the model's prediction.

In Figure 9, something even more interesting occurs: the two corpus decompositions suggest different conclusions. In the American doctor's perspective, the prediction for this patient appears perfectly coherent as all patients in the corpus decomposition have very similar features and all of them are rightfully classified. On the other hand, the British doctor will reach the opposite conclusion as the most relevant corpus patient is misclassified by the model. In this case, we have a perfect illustration of the limitation of the transfer of a risk model from one site (America) to another (United Kingdom): similar patients from different sites can have different outcomes. In both cases, since the test patient is British, only the decomposition in terms of $\mathcal{C}_{\mathrm{UK}}$ really matters. In both cases, the British doctor could have anticipated the misclassification of each patient with SimplEx.

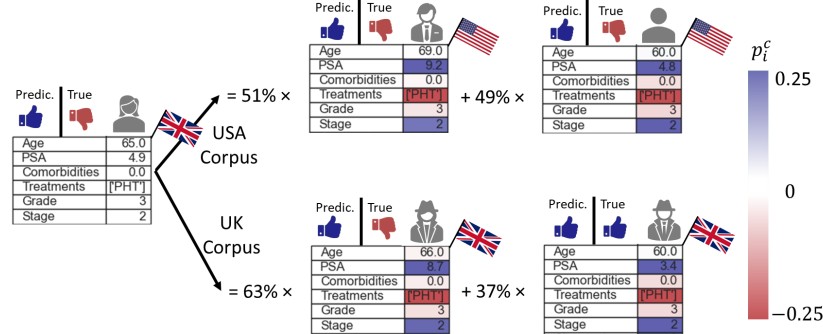

Figure 8: A first misclassified patient.

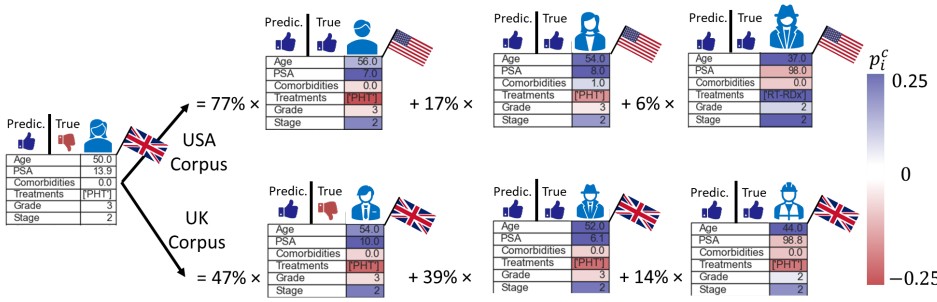

Figure 9: A second misclassified patient.

## 4 Discussion

We have introduced SimplEx, a method that decomposes the model representations at inference time in terms of a corpus. Through several experiments, we have demonstrated that these decompositions are accurate and can easily be personalized to the user. Finally, by introducing Integrated Jacobians, we have brought these explanations to the feature level.

We believe that our bridge between feature and example-based explainability opens up many avenues for the future. A first interesting extension would be to investigate how SimplEx can be used to understand latent representations involved in unsupervised learning. For instance, SimplEx could be used to study the interpretability of *self-expressive* latent representations learned by autoencoders [35]. A second interesting possibility would be to design a rigorous scheme to select the optimal corpus for a given model and dataset. Finally, a formulation where we allow the corpus to vary on the basis of observations would be particularly interesting for online learning.

## Acknowledgement

The authors are grateful to Alicia Curth, Krzysztof Kacprzyk, Boris van Breugel, Yao Zhang and the 4 anonymous NeurIPS 2021 reviewers for their useful comments on an earlier version of the manuscript. Jonathan Crabbé would like to thank Bogdan Cebere for replicating the experiments in this manuscript. Jonathan Crabbé would like to acknowledge Bilyana Tomova for many insightful discussions and her constant support. Jonathan Crabbé is funded by Aviva. Fergus Imrie is supported by by the National Science Foundation (NSF), grant number 1722516. Zhaozhi Qian and Mihaela van der Schaar are supported by the Office of Naval Research (ONR), NSF 1722516.

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
