# Supplementary Material: Explaining Latent Representations with a Corpus of Examples

**Jonathan Crabbé**
University of Cambridge
jc2133@cam.ac.uk

**Zhaozhi Qian**
University of Cambridge
zq224@maths.cam.ac.uk

**Fergus Imrie**
UCLA
imrie@g.ucla.edu

**Mihaela van der Schaar**
University of Cambridge
The Alan Turing Institute
UCLA
mv472@cam.ac.uk

## 1 Supplement for Mathematical Formulation

In this supplementary section, we give more details on the mathematical aspects underlying SimplEx.

### 1.1 Precision of the corpus approximation in output space

If the corpus representation of $\mathbf{h} \in \mathcal{H}$ has a residual $r_{\mathcal{C}}(\mathbf{h})$, Assumption 2.1 controls the error between the black-box prediction for the test example $\mathbf{f}(\mathbf{x}) = \mathbf{l}(\mathbf{h})$ and and its corpus representation $\mathbf{l}(\hat{\mathbf{h}})$.

**Proposition 1.1** (Precision in output space). *Consider a latent representation $\boldsymbol{h}$ with corpus residual $r_{\mathcal{C}}(\boldsymbol{h})$. If Assumption 2.1 holds, this implies that the corpus prediction $\boldsymbol{l}(\hat{\boldsymbol{h}})$ approximates $\boldsymbol{l}(\boldsymbol{h})$ with a precision controlled by the corpus residual:*

$$\| \, \boldsymbol{l}(\hat{\boldsymbol{h}}) - \boldsymbol{l}(\boldsymbol{h}) \, \|_{\mathcal{Y}} \quad \leq \quad \| \, \boldsymbol{l} \, \|_{op} \cdot r_{\mathcal{C}}(\boldsymbol{h}),$$

*where $\| \cdot \|_{\mathcal{Y}}$ is a norm on $\mathcal{Y}$ and $\| \, \boldsymbol{l} \, \|_{op} = \inf \left\{ \lambda \in \mathbb{R}^+ : \| \, \boldsymbol{l}(\tilde{\boldsymbol{h}}) \, \|_{\mathcal{Y}} \leq \lambda \, \| \, \tilde{\boldsymbol{h}} \, \|_{\mathcal{H}} \ \ \forall \tilde{\boldsymbol{h}} \in \mathcal{H} \right\}$ is the usual operator norm.*

*Proof.* The proof is immediate:

$$
\begin{aligned}
\| \, \mathbf{l}(\hat{\mathbf{h}}) - \mathbf{l}(\mathbf{h}) \, \|_{\mathcal{Y}} \quad &= \quad \| \, \mathbf{l}(\hat{\mathbf{h}} - \mathbf{h}) \, \|_{\mathcal{Y}} \\
&\leq \quad \| \, \mathbf{l} \, \|_{\mathrm{op}} \cdot \| \, \hat{\mathbf{h}} - \mathbf{h} \, \|_{\mathcal{H}} \\
&= \quad \| \, \mathbf{l} \, \|_{\mathrm{op}} \cdot r_{\mathcal{C}}(\mathbf{h}),
\end{aligned}
$$

where we have successively used the linearity of $\mathbf{l}$, the definition of the operator norm $\| \cdot \|_{\mathrm{op}}$ and Definition 2.2. $\qquad \square$

### 1.2 Uniqueness of corpus decomposition

As we have mentioned in the main paper, the corpus decomposition is not always unique. To illustrate, we consider the following corpus representation: $\mathbf{g}(\mathcal{C}) = \left\{ \mathbf{h}^1, \mathbf{h}^2, \mathbf{h}^3 = 0.5 \cdot \mathbf{h}^1 + 0.5 \cdot \mathbf{h}^2 \right\}$. Consider the following vector in the corpus hull: $\mathbf{h} = 0.75 \cdot \mathbf{h}^1 + 0.25 \cdot \mathbf{h}^2$. We note that this vector can also be written as $\mathbf{h} = 0.5 \cdot \mathbf{h}^1 + 0.5 \cdot \mathbf{h}^3$. In other words, the vector $\mathbf{h} \in \mathbf{g}(\mathcal{C})$ admits more than one corpus decomposition. This is not a surprise for the attentive reader: by paying a closer

35th Conference on Neural Information Processing Systems (NeurIPS 2021).

look to $\mathbf{g}(\mathcal{C})$, we note that $\mathbf{h}^3$ is somewhat redundant as it is itself a combination of $\mathbf{h}^1$ and $\mathbf{h}^2$. The multiplicity of the corpus decomposition results from a redundancy in the corpus representation.

To make this reasoning more general, we need to revisit some classic concepts of convex analysis. To establish a sufficient condition that guarantees the uniqueness of corpus decompositions, we recall the definition of affine independence.

**Definition 1.1** (Affine independence). The vectors $\{\mathbf{h}^c \mid c \in [C]\} \subset \mathbb{R}^d$ are *affinely independent* if

$$\sum_{c=1}^{C} \lambda^c \mathbf{h}^c = 0 \ \wedge \ \sum_{c=1}^{C} \lambda^c = 0 \implies \lambda^c = 0 \ \forall \ c \in [C]$$

If a set of vectors is not affinely independent, it means that one of the vectors can be written as an affine combination of the others. This is precisely what we called a redundancy in the previous paragraph. We now adapt a well-known result of convex analysis to our formalism:

**Proposition 1.2** (Uniqueness of corpus decomposition). *If the corpus representation $\mathbf{g}(\mathcal{C}) = \{\mathbf{h}^c \mid c \in [C]\}$ is a set of affinely independent vectors, then every vector in the corpus hull $\mathbf{h} \in \mathcal{CH}(\mathcal{C})$ admits one unique corpus decomposition.*

*Proof.* The existence of a decomposition is a trivial consequence of the definition of $\mathcal{CH}(\mathcal{C})$. We prove the uniqueness of the decomposition by contradiction. Let us assume that a vector $\mathbf{h} \in \mathcal{CH}(\mathcal{C})$ admits two distinct corpus decompositions:

$$\mathbf{h} = \sum_{c=1}^{C} w^c \mathbf{h}^c = \sum_{c=1}^{C} \tilde{w}^c \mathbf{h}^c$$

where $w^c, \tilde{w}^c \geq 0$ for all $c \in [C]$, $\sum_{c=1}^{C} w^c = \sum_{c=1}^{C} \tilde{w}^c = 1$ and $(w^c)_{c=1}^{C} \neq (\tilde{w}^c)_{c=1}^{C}$. It follows that:

$$\sum_{c=1}^{C} \underbrace{(w^c - \tilde{w}^c)}_{\equiv \lambda^c} \mathbf{h}^c = 0$$

But $\sum_{c=1}^{C} \lambda^c = \sum_{c=1}^{C} (w^c - \tilde{w}^c) = 1 - 1 = 0$. It follows that $\mathbf{g}(\mathcal{C})$ is not affinely independent, a contradiction. $\square$

This shows that affine independence provides a sufficient condition to ensure the uniqueness of corpus decompositions. If one wants to produce such a corpus, a possibility is to gradually add new examples in the corpus by checking that the latent representation of each new example is not an affine combination of the previous latent representations. Clearly, the number of examples in such a corpus cannot exceed $d_H + 1$.

### 1.3 Integrated Jacobian and Integrated Gradients

Integrated Gradients is a notorious method used to discuss feature saliency [1]. It uses a black-box output to attribute a saliency score to each feature. In the original paper, the output space $\mathcal{Y}$ is assumed to be one-dimensional: $d_Y = 1$. We shall therefore relax the bold notation that we have used for the outputs so far. In this way, the black-box is denoted $f$ and the latent-to-output map is denoted $l$. Although the original paper makes no mention of corpus decompositions, it is straightforward to adapt the definition of Integrated Gradients to our set-up:

**Definition 1.2** (Integrated Gradient). The *Integrated Gradient* between a baseline $\mathbf{x}^0$ an a corpus example $\mathbf{x}^c \in \mathcal{X}$ associated to feature $i \in [d_X]$ is

$$\mathrm{IG}_i^c = \left( x_i^c - x_i^0 \right) \int_0^1 \left. \frac{\partial f}{\partial x_i} \right|_{\boldsymbol{\gamma}^c(t)} dt \quad \in \mathbb{R},$$

where $\boldsymbol{\gamma}^c(t) \equiv \mathbf{x}^0 + t \cdot \left( \mathbf{x}^c - \mathbf{x}^0 \right)$ for $t \in [0, 1]$.

In the main paper, we have introduced Integrated Jacobians: a latent space generalization of Integrated Gradients. We use the word generalization for a reason: the Integrated Gradient can be deduced from the Integrated Jacobian but not the opposite[1]. We make the relationship between the two quantities explicit in the following proposition.

**Proposition 1.3.** *The Integrated Gradient can be deduced from the Integrated Jacobian via*

$$IG_i^c = l\left(\boldsymbol{j}_i^c\right).$$

*Proof.* We start from the definition of the Integrated Gradient:

$$
\begin{aligned}
\text{IG}_i^c &= \left(x_i^c - x_i^0\right) \int_0^1 \left.\frac{\partial f}{\partial x_i}\right|_{\boldsymbol{\gamma}^c(t)} dt \\
&= \left(x_i^c - x_i^0\right) \int_0^1 \left.\frac{\partial (l \circ \mathbf{g})}{\partial x_i}\right|_{\boldsymbol{\gamma}^c(t)} dt \\
&= \left(x_i^c - x_i^0\right) \int_0^1 l\left(\left.\frac{\partial \mathbf{g}}{\partial x_i}\right|_{\boldsymbol{\gamma}^c(t)}\right) dt \\
&= \left(x_i^c - x_i^0\right) l\left(\int_0^1 \left.\frac{\partial \mathbf{g}}{\partial x_i}\right|_{\boldsymbol{\gamma}^c(t)} dt\right) \\
&= l\left(\left(x_i^c - x_i^0\right) \int_0^1 \left.\frac{\partial \mathbf{g}}{\partial x_i}\right|_{\boldsymbol{\gamma}^c(t)} dt\right) \\
&= l\left(\mathbf{j}_i^c\right),
\end{aligned}
$$

where we have successively used: Assumption 2.1, the linearity of the partial derivative, the linearity of the integration operator, the linearity of $l$ and the definition of Integrated Jacobians. $\square$

Note that Integrated Jacobians allow us to push our understanding of the black-box beyond the output. There is very little reason to expect a one dimensional output to capture the model complexity. As we have argued in the introduction, our paper pursues the more challenging ambition of gaining a deeper understanding of the black-box latent space.

## 1.4 Properties of Integrated Jacobians

We give a proof for the proposition appearing in the main paper.

**Proposition 2.1** (Properties of Integrated Jacobians). *Consider a baseline $(\boldsymbol{x}^0, \boldsymbol{h}^0 = \boldsymbol{g}(\boldsymbol{x}^0))$ and a test example together with their latent representation $(\boldsymbol{x}, \boldsymbol{h} = \boldsymbol{g}(\boldsymbol{x})) \in \mathcal{X} \times \mathcal{H}$. If the shift $\boldsymbol{h} - \boldsymbol{h}^0$ admits a decomposition (2), the following properties hold.*

$$(A): \quad \sum_{c=1}^{C} \sum_{i=1}^{d_X} w^c \boldsymbol{j}_i^c = \boldsymbol{h} - \boldsymbol{h}^0 \qquad (B): \quad \sum_{c=1}^{C} \sum_{i=1}^{d_X} w^c p_i^c = 1.$$

*Proof.* Let us begin by proving (A). By using the chain rule for a given corpus example $c \in [C]$, we write explicitly the derivative of the curve $\mathbf{g} \circ \boldsymbol{\gamma}^c$ with respect to its parameter $t \in (0, 1)$:

$$
\begin{aligned}
\left.\frac{d\left(\mathbf{g} \circ \boldsymbol{\gamma}^c\right)}{dt}\right|_t &= \sum_{i=1}^{d_X} \left.\frac{\partial \mathbf{g}}{\partial x_i}\right|_{\boldsymbol{\gamma}^c(t)} \left.\frac{d\gamma_i^c}{dt}\right|_t \\
&= \sum_{i=1}^{d_X} \left.\frac{\partial \mathbf{g}}{\partial x_i}\right|_{\boldsymbol{\gamma}^c(t)} (x_i^c - x_i^0),
\end{aligned}
$$

---

[1]Unless in the degenerate case where $d_H = 1$. However, this case is of little interest as it describes a situation where $\mathcal{Y}$ and $\mathcal{H}$ are isomorphic, hence the distinction between output and latent space is fictional.

where we used $\gamma_i^c(t) = x_i^0 + t \cdot (x_i^c - x_i^0)$ to obtain the second equality. We use this equation to rewrite the sum of the Integrated Jacobians for this corpus example $c$:

$$
\begin{aligned}
\sum_{i=1}^{d_X} \mathbf{j}_i^c &= \sum_{i=1}^{d_X} \int_0^1 \left.\frac{\partial \mathbf{g}}{\partial x_i}\right|_{\boldsymbol{\gamma}^c(t)} (x_i^c - x_i^0)\, dt \\
&= \int_0^1 \sum_{i=1}^{d_X} \left.\frac{\partial \mathbf{g}}{\partial x_i}\right|_{\boldsymbol{\gamma}^c(t)} (x_i^c - x_i^0)\, dt \\
&= \int_0^1 \left.\frac{d\,(\mathbf{g} \circ \boldsymbol{\gamma}^c)}{dt}\right|_t dt \\
&= \mathbf{g} \circ \boldsymbol{\gamma}^c(1) - \mathbf{g} \circ \boldsymbol{\gamma}^c(0) \\
&= \mathbf{h}^c - \mathbf{h}^0,
\end{aligned}
$$

where we have successively used: the linearity of integration, the explicit expression for the curve derivative, the fundamental theorem of calculus and the definition of the curve $\mathbf{g} \circ \boldsymbol{\gamma}^c$. We are now ready to derive (A):

$$
\begin{aligned}
\sum_{c=1}^{C} \sum_{i=1}^{d_X} w^c \mathbf{j}_i^c &= \sum_{c=1}^{C} w^c \left(\mathbf{h}^c - \mathbf{h}^0\right) \\
&= \mathbf{h} - \mathbf{h}^0,
\end{aligned}
$$

where we have successively used the exact expression for the sum of Integrated Jacobians associated to corpus example $c$ and the definition of the corpus decomposition of $\mathbf{h}$. We are done with (A), let us now prove (B). We simply project both members of (A) on the overall shift $\mathbf{h} - \mathbf{h}^0$. Projecting the left-hand side of (A) yields:

$$
\operatorname{proj}_{\mathbf{h}-\mathbf{h}^0}\left(\sum_{c=1}^{C} \sum_{i=1}^{d_X} w^c \mathbf{j}_i^c\right) = \sum_{c=1}^{C} \sum_{i=1}^{d_X} w^c \underbrace{\operatorname{proj}_{\mathbf{h}-\mathbf{h}^0}\left(\mathbf{j}_i^c\right)}_{p_i^c},
$$

where we used the linearity of the projection operator. Projecting the right-hand side of (A) yields:

$$
\operatorname{proj}_{\mathbf{h}-\mathbf{h}^0}(\mathbf{h} - \mathbf{h}^0) = \frac{\langle\, \mathbf{h} - \mathbf{h}^0 \,,\, \mathbf{h} - \mathbf{h}^0 \,\rangle}{\langle\, \mathbf{h} - \mathbf{h}^0 \,,\, \mathbf{h} - \mathbf{h}^0 \,\rangle} = 1.
$$

By equating the projected version of both members of (A), we deduce (B). $\qquad\square$

### 1.5 Pseudocode for SimplEx

We give the pseudocode for the two modules underlying SimplEx: the corpus decomposition (Algorithm 1) and the evaluation of projected Jacobians (Algorithm 2).

---

**Algorithm 1:** SimplEx: Corpus Decomposition

---

**Input:** Test latent representation $\mathbf{h}$ ; Corpus representation $\{\mathbf{h}^c \mid c \in [C]\}$
**Result:** Weights of the corpus decomposition $\mathbf{w} \in [0,1]^C$ ; Corpus residual $r_{\mathcal{C}}(\mathbf{h})$
Initialize pre-weights: $\tilde{\mathbf{w}} \leftarrow \mathbf{0}$;
**while** *optimizing* **do**

    Normalize pre-weights: $\mathbf{w} \leftarrow \mathbf{softmax}\,[\tilde{\mathbf{w}}]$;

    Evaluate loss: $L\,[\tilde{\mathbf{w}}] \leftarrow \sum_{i=1}^{d_H}\left(h_i - \sum_{c=1}^{C} w^c h_i^c\right)^2$;

    Update pre-weights: $\tilde{\mathbf{w}} \leftarrow \mathbf{Adam\ step}\,(L\,[\tilde{\mathbf{w}}])$;

**end**
Return normalized weights: $\mathbf{w} \leftarrow \mathbf{softmax}\,[\tilde{\mathbf{w}}]$;

Return corpus residual: $r_{\mathcal{C}}(\mathbf{h}) \leftarrow \left[\sum_{i=1}^{d_H}\left(h_i - \sum_{c=1}^{C} w^c h_i^c\right)^2\right]^{1/2}$;

---

Where we used a vector notation for the pre-weights and the weights: $\mathbf{w} = (w^c)_{c=1}^C$. For the Adam optimizer, we use the default hyperparameters in the Pytorch implementation: $\mathrm{lr} = 10^{-3}$ ; $\beta_1 = .9$ ; $\beta_2 = .999$ ; eps $= 10^{-8}$. Note that this algorithm is a standard optimization loop for a convex problem where the normalization of the weights is ensured by using a softmax.

When the size of the corpus elements to contribute has to be limited to $K$, we use a similar strategy as the one used to produce extremal perturbations [2, 3]. This consists in adding the following $L^1$ term to the optimized loss $L$:

$$L_{\mathrm{reg}}[\tilde{\mathbf{w}}] = \sum_{d=1}^{C-K} \left| \mathrm{vecsort}^d[\tilde{\mathbf{w}}] \right|,$$

where vecsort is a permutation operator that sorts the components of a vector in ascending order. The notation $\mathrm{vecsort}^d$ refers to the $d^{th}$ component of the sorted vector. This regularization term will impose sparsity for the $C - K$ smallest weights of the corpus decomposition. As a result, the optimal corpus decomposition only involves $K$ non-vanishing weights. We now focus on the evaluation of the Projected Jacobian.

---

**Algorithm 2:** SimplEx: Projected Jacobian

---

**Input:** Test input $\mathbf{x}$ ; Test representation $\mathbf{h}$ ; Corpus $\{\mathbf{x}^c \mid c \in [C]\}$ ; Corpus representation $\{\mathbf{h}^c \mid c \in [C]\}$ ; Baseline input $\mathbf{x}^0$ ; Baseline representation $\mathbf{h}^0$ ; Black-box latent map $\mathbf{g}$ ; Number of bins $N_b \in \mathbb{N}^*$

**Result:** Jacobian projections $\mathbf{P} = (p_i^c) \in \mathbb{R}^{C \times d_X}$

Initialize the projection matrix: $\mathbf{P} = (p_i^c) \leftarrow \mathbf{0}$ ;

Form a matrix of corpus inputs: $\mathbf{X}^C \leftarrow (x_i^c) \in \mathbb{R}^{C \times d_X}$ ;

Form a matrix of baseline inputs: $\mathbf{X}^0 \leftarrow (x_i^0) \in \mathbb{R}^{C \times d_X}$ ;

**for** $n \in [N_b]$ **do**

    Set the evaluation input: $\tilde{\mathbf{X}} \leftarrow \mathbf{X}^0 + \frac{n}{N_b}\left(\mathbf{X}^C - \mathbf{X}^0\right)$ ;

    Increment the Jacobian projections: $p_i^c \leftarrow p_i^c + \frac{\partial \mathbf{g}}{\partial x_i}\big|_{\tilde{\mathbf{x}}^c} \cdot \frac{\mathbf{h} - \mathbf{h}^0}{\|\mathbf{h} - \mathbf{h}^0\|_2^2} \quad \forall (c, i) \in [C] \times [d_X]$ ;

**end**

Apply the appropriate pre-factor: $\mathbf{P} \leftarrow \frac{1}{N_b}\left(\mathbf{X}^C - \mathbf{X}^0\right) \odot \mathbf{P}$ ;

---

This algorithm approximates the integral involved in the definition of the Projected Jacobian with a standard Riemann sum. Note that the definition of $\mathbf{X}^0$ implies that the baseline vector $\mathbf{x}^0$ is broadcasted along the first dimension of the matrix. More explicitly, the components of this matrix are $X_{c,i}^0 = x_i^0$ for $c \in [C]$ and $i \in [d_X]$. Also note that the projected Jacobians can be computed in parallel with packages such as Pytorch's autograd. We have used the notation $\odot$ to denote the conventional Hadarmard product. In our implementation, the number of bins $N_b$ is fixed at 200, bigger $N_b$ don't significantly improve the precision of the Riemann sum.

## 1.6 Choice of a baseline for Integrated Jacobians

Throughout our analysis of the corpus decomposition, we have assumed the existence of a baseline $\mathbf{x}_0 \in \mathcal{X}$. This baseline is crucial as it defines the starting point of the line $\gamma^c$ that we use to compute the Jacobian quantities. What is a good choice for the baseline? The answer to this question depends on the domain. When this makes sense, we choose the baseline to be an instance that does not contain any information. A good example of this is the baseline that we use for MNIST: an image that is completely black $\mathbf{x}_0 = 0$. Sometimes, this absence of information is not well-defined. A good example is the prostate cancer experiment: it makes little sense to define a patient whose features contain no information. In this set-up, our baseline is a patient whose features are fixed to their average value in the training [2] set $\mathbf{x}_0 = |\mathcal{D}_{\mathrm{train}}|^{-1} \sum_{\mathbf{x} \in \mathcal{D}_{\mathrm{train}}} \mathbf{x}$. In this way, a shift with respect to the baseline corresponds to a patient whose features differ from the population average.

---

[2] This average could also be computed with respect to the corpus itself.

| Feature | Range |
|---|---|
| Age | $60 - 73$ |
| PSA | $5 - 11$ |
| Comorbidities | $0, 1, 2, \geq 3$ |
| Treatment | Hormone Therapy (PHT), Radical Therapy - RDx (RT-RDx), Radical Therapy -Sx (RT-Sx), CM |
| Grade | $1, 2, 3, 4, 5$ |
| Stage | $1, 2, 3, 4$ |
| Primary Gleason | $1, 2, 3, 4, 5$ |
| Secondary Gleason | $1, 2, 3, 4, 5$ |

Table 1: Features for the SEER Dataset.

## 2 Supplement for Experiments

In this section, we give more details on the experiments that we have conducted with SimplEx. All our experiments have been performed on a machine with Intel(R) Core(TM) i5-8600K CPU @ 3.60GHz [6 cores] and Nvidia GeForce RTX 2080 Ti GPU. Our implementation is done with Pytorch 1.8.1.

### 2.1 Details for the corpus precision experiment

**Metrics** We give the explicit expression for the two metrics used in the experiment. By keeping the notation of Section 3, we assume that we are given a set of test samples $\mathcal{T} \subset \mathcal{X}$. For each test representation $\mathbf{h} \in \mathbf{g}(\mathcal{T})$ and test output $\mathbf{y} \in \mathbf{f}(\mathcal{T})$, we build build an approximation $\hat{\mathbf{h}}$ and $\hat{\mathbf{y}}$ with several methods. To evaluate the quality of these approximations, we use the following $R^2$ scores:

$$R_{\mathcal{H}}^2 = 1 - \frac{\sum_{\mathbf{h} \in \mathbf{g}(\mathcal{T})} \| \mathbf{h} - \hat{\mathbf{h}} \|_2^2}{\sum_{\mathbf{h} \in \mathbf{g}(\mathcal{T})} \| \mathbf{h} - \bar{\mathbf{h}} \|_2^2} \qquad \bar{\mathbf{h}} = \frac{1}{|\mathcal{T}|} \sum_{\mathbf{h} \in \mathbf{g}(\mathcal{T})} \mathbf{h}$$

$$R_{\mathcal{Y}}^2 = 1 - \frac{\sum_{\mathbf{y} \in \mathbf{f}(\mathcal{T})} \| \mathbf{y} - \hat{\mathbf{y}} \|_2^2}{\sum_{\mathbf{y} \in \mathbf{f}(\mathcal{T})} \| \mathbf{y} - \bar{\mathbf{y}} \|_2^2} \qquad \bar{\mathbf{y}} = \frac{1}{|\mathcal{T}|} \sum_{\mathbf{y} \in \mathbf{f}(\mathcal{T})} \mathbf{y}.$$

These $R^2$ scores compare the approximation method to a dummy approximator that approximates every representation and output with their test average. A negative value for $R^2$ indicates that the approximation method performs more poorly than the dummy approximator. Ideally, the $R^2$ score should be close to 1.

**SEER Dataset** The SEER dataset is a private dataset consisting in 240,486 patients enrolled in the American SEER program [4]. All the patients from the SEER dataset have been de-identified. We consider the binary classification task of predicting cancer mortality for patients with prostate cancer. Each patient in the dataset is represented by the couple $(\mathbf{x}, \mathbf{z})$, where $\mathbf{x}$ contains the patient features and $\mathbf{z} \in \{0, 1\}^2$ is a vector indicating the patient mortality. The features characterizing each patient are their age, PSA, Gleason score, clinical stage and which, if any, treatment they are receiving. These features are summarized in Table 1. The original dataset is severely imbalanced as $93.8\%$ patients survive or have a mortality unrelated to cancer. We extract a balanced subset of 42,000 patients that we split into a training set $\mathcal{D}_{\text{train}}$ of 35,700 patients and a test set $\mathcal{D}_{\text{test}}$ of 6,300 patients. We train a multilayer perceptron (MLP) for the mortality prediction task on $\mathcal{D}_{\text{train}}$.

**MNIST Dataset** MNIST is a public dataset consisting in 70,000 MNIST images of handwritten digits [5]. We consider the multiclass classification task of identifying the digit represented on each image. Each instance in the dataset is represented by the couple $(\mathbf{x}, \mathbf{z})$, where $\mathbf{x}$ contains the image itself and $\mathbf{z} \in \{0, 1\}^{10}$ is a vector indicating the true label for the image. The images are characterized by $28 \times 28$ pixels with one channel. The dataset is conventionally split into a training set $\mathcal{D}_{\text{train}}$ of 60,000 images and a test set $\mathcal{D}_{\text{test}}$ of 10,000 images. We train a convolutional neural network (CNN) for the image classification task on $\mathcal{D}_{\text{train}}$. Yann LeCun and Corinna Cortes hold the copyright of MNIST dataset, which is a derivative work from original NIST datasets. MNIST dataset is made available under the terms of the Creative Commons Attribution-Share Alike 3.0 license.

| Layer | Input Dimension | Output Dimension | Activation | Remark |
|---|---|---|---|---|
| Batch Norm | 3 | 3 | | Only acts on Age, PSA and Comorbidities |
| Dense 1 | 26 | 200 | ReLU | |
| Dropout | 200 | 200 | | |
| Dense 2 | 200 | 50 | ReLU | |
| Dropout | 50 | 50 | | Output: $\mathbf{h} = \mathbf{g}(\mathbf{x})$ |
| Linear | 50 | 2 | | Output: $\mathbf{y} = \mathbf{l}(\mathbf{h})$ |
| Softmax | 2 | 2 | | Output: $\mathbf{p} = \mathbf{f}(\mathbf{x})$ |

Table 2: Mortality Prediction MLP for SEER.

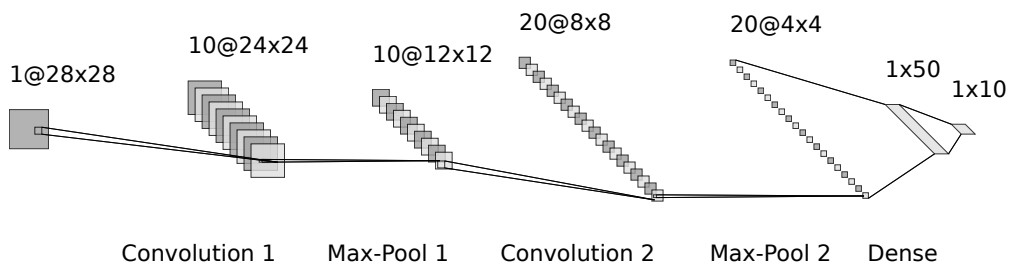

Figure 1: Architecture of MNIST classifier.

**Prostate Cancer Model** The model that we use for mortality prediction on the SEER dataset is a simple MLP with two hidden layers. Its precise architecture is described in Table 2. The model is trained by minimizing the following loss:

$$\mathcal{L}_{\text{train}}(\boldsymbol{\theta}) = \sum_{(\mathbf{x},\mathbf{z}) \in \mathcal{D}_{\text{train}}} -\mathbf{z} \odot \log[\mathbf{f}_{\boldsymbol{\theta}}(\mathbf{x})] + \lambda \cdot \| \boldsymbol{\theta} \|_2, \tag{1}$$

where we have introduced a $L^2$ regularization, as required by the representer theorem. The regularization coefficient is chosen to be $\lambda = 10^{-5}$ (a bigger $\lambda$ significantly decreases the performance of the model on the testing set). We train this model with Adam (default Pytorch hyperparameters) for 5 epochs. Across the different runs, the accuracy of the resulting model on the test set ranges between $85 - 86\%$.

**MNIST Model** The model that we use for image classification on the MNIST dataset is the CNN represented in Figure 1. Its precise architecture is described in Table 3. The model is trained by minimizing the loss (1) with $\lambda = 10^{-1}$. We train this model with Adam (default Pytorch hyperparameters) for 10 epochs. Across the different runs, the accuracy of the resulting model on the test set ranges between $94 - 96\%$ (note that the weight decay decreases the performances, training the same model with $\lambda = 0$ yields a test accuracy above $99\%$).

**Representer theorem** Previous works established that the pre-activation output of classification deep-networks can be decomposed in terms of contributions arising from the training set [6]. In our set-up, where the neural network takes the form $\mathbf{f} = \phi \circ \mathbf{l} \circ \mathbf{g}$, the decomposition can be written as

$$\mathbf{y} = \mathbf{l} \circ \mathbf{g}(\mathbf{x})$$
$$= -\frac{1}{2\lambda|\mathcal{D}_{\text{train}}|} \sum_{(\mathbf{x}',\mathbf{z}') \in \mathcal{D}_{\text{train}}} \frac{\partial \mathcal{L}_{\text{train}}}{\partial [\mathbf{l}(\mathbf{x}')]}$$
$$= \frac{1}{2\lambda|\mathcal{D}_{\text{train}}|} \sum_{(\mathbf{x}',\mathbf{z}') \in \mathcal{D}_{\text{train}}} [\mathbf{z}' - \mathbf{f}(\mathbf{x}')] \cdot [\mathbf{g}(\mathbf{x}')]^\top [\mathbf{g}(\mathbf{x})],$$

| Layer | Input Dimension | Output Dimension | Activation | Remark |
|---|---|---|---|---|
| Convolution 1 | $28 \times 28 \times 1$ | $24 \times 24 \times 10$ | | Kernel Size: 5 |
| Max-Pool 1 | $24 \times 24 \times 10$ | $12 \times 12 \times 10$ | ReLU | Kernel Size: 2 |
| Convolution 2 | $12 \times 12 \times 10$ | $8 \times 8 \times 20$ | | Kernel Size: 5 |
| Dropout | $8 \times 8 \times 20$ | $8 \times 8 \times 20$ | | |
| Max-Pool 2 | $8 \times 8 \times 20$ | $4 \times 4 \times 20$ | ReLU | Kernel Size: 2 |
| Flatten | $4 \times 4 \times 20$ | 320 | | |
| Dense | 320 | 50 | ReLU | |
| Dropout | 50 | 50 | | Output: $\mathbf{h} = \mathbf{g}(\mathbf{x})$ |
| Linear | 50 | 10 | | Output: $\mathbf{y} = \mathbf{l}(\mathbf{h})$ |
| Softmax | 10 | 10 | | Output: $\mathbf{p} = \mathbf{f}(\mathbf{x})$ |

Table 3: MNIST Classifier CNN.

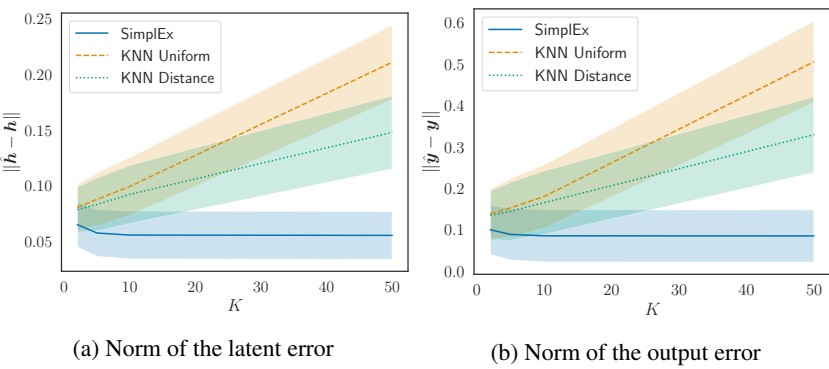

(a) Norm of the latent error

(b) Norm of the output error

Figure 2: Precision of corpus decomposition for prostate cancer (avg $\pm$ std).

where $\lambda$ is the $L^2$ regularization coefficient used in the training loss (1). In our work, we decompose the same output in terms of a corpus $\mathcal{C}$ that is distinct from the training set $\mathcal{D}_{\text{train}}$. In this experiment, the corpus is a random subset of the training set $\mathcal{C} \subset \mathcal{D}_{\text{train}}$ (in our implementation of the representer theorem, the true labels associated to the corpus example are included). To give a corpus approximation of the output with the representer theorem, we restrict the above sum to the corpus:

$$\hat{\mathbf{y}} = \frac{1}{2\lambda|\mathcal{C}|} \sum_{(\mathbf{x}',\mathbf{z}')\in\mathcal{C}} [\mathbf{z}' - \mathbf{f}(\mathbf{x}')] \cdot [\mathbf{g}(\mathbf{x}')]^\top [\mathbf{g}(\mathbf{x})] .$$

The $R_{\mathcal{Y}}^2$ score reported in the main paper measure the quality of this approximation. It turns out that $R_{\mathcal{Y}}^2 < 0$ in both experiments, which indicates that the representer theorem offers poor approximations. We have two explanations: (1) As previously mentioned, the representer theorem assumes that the decomposition involves the *whole* training set. By making a decomposition that involves a subset $\mathcal{C}$ of the training set, we violate a first assumption of the representer theorem. (2) The representer theorem assumes that the trained model $\mathbf{f}_{\boldsymbol{\theta}^*}$ corresponds to a stationnary point of the loss: $\nabla_{\boldsymbol{\theta}} \mathcal{L}_{\text{train}} |_{\boldsymbol{\theta}^*} = 0$. This assumption is rarely verified in non-convex optimization problems such as the optimization of deep networks.

**Plots with an alternative metric** In Figures 2 & 3, we report the norm of the error associated to each method as a function of the number of active corpus members $K$. With this metric, SimplEx remains the most interesting approximation method in latent and output space. Note that for SimplEx, when $K = C$, the error in latent space is equivalent to the corpus residual $r_{\mathcal{C}}(\mathbf{h})$.

**On the measure of consistency** In our experiments, we use the standard deviation of each metric across different runs to study the consistency of the corpus approximations. Table 4 details the parts of the experiment that are modified from one run to another.

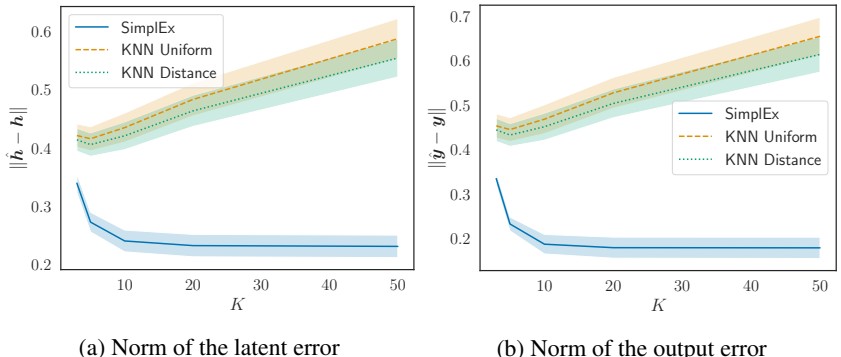

(a) Norm of the latent error        (b) Norm of the output error

Figure 3: Precision of corpus decomposition for MNIST (avg $\pm$ std).

| Modified on each run | Dataset | | |
| --- | --- | --- | --- |
| | Prost. Cancer | MNIST | AR |
| Data | | | ✓ |
| Train-Test Split | ✓ | | ✓ |
| Model | ✓ | ✓ | ✓ |
| Corpus | ✓ | ✓ | ✓ |

Table 4: Characterization of a run for each dataset.

## 2.2 Details for the clinical use case

**Description** In this subsection, we detail the detection of British patients that is described in Section 3.3 of the main paper.

**Metrics** Each method produces an ordered list $(\mathbf{x}^m)_{m=1}^{|\mathcal{T}|}$ of elements of $\mathcal{T}$. We inspect its elements in order and count the number of examples that are in $\mathcal{T} \cap \mathcal{D}_{\mathrm{UK}}$. This count is represented by the sequence $(u_n)_{n=1}^{|\mathcal{T}|}$ where $u_n = |(\mathbf{x}^m)_{m=1}^n \bigcap \mathcal{D}_{\mathrm{UK}}|$. A sequence $(u_n)_{n=1}^{|\mathcal{T}|}$ that increases more quickly is better as it corresponds to a more efficient detection of the British patients. The maximal baseline corresponds to the upper bound. The experiments are repeated 10 times to report the average metrics together with their standard deviations.

**Baselines** We compare SimplEx with 5 indicative baselines. Each method produces an ordered list $(\mathbf{x}^m)_{m=1}^{|\mathcal{T}|}$ of elements of $\mathcal{T}$. In the case of SimplEx, this list is produced by sorting the examples in decreasing order of corpus residual. We use the two Nearest Neighbours baselines from the previous examples by fixing $K$ to the value that produced the best approximations ($K = 7$). For both of these baselines, we sort the examples in decreasing order of residual $\|\mathbf{h} - \hat{\mathbf{h}}\|$. We consider the random baseline where the order of the list is chosen randomly. Finally, we introduce the ideal baseline that detects all outliers with $|\mathcal{T}|/2 = 100$ inspections: $(\mathbf{x}^m)_{m=1}^{|\mathcal{T}|/2} = \mathcal{T} \cap \mathcal{D}_{\mathrm{out}}$.

**CUTRACT Dataset** The CUTRACT dataset is a private dataset consisting in 10,086 patients enrolled in the British Prostate Cancer UK program [7]. All the patients from the CUTRACT dataset have been de-identified. We consider the binary classification task of predicting cancer mortality for patients with prostate cancer. Each patient in the dataset is represented by the couple $(\mathbf{x}, \mathbf{z})$, where $\mathbf{x}$ contains the patient features and $\mathbf{z} \in \{0, 1\}^2$ is a vector indicating the patient mortality. The features characterizing the patient are the same as for the SEER dataset. These features are summarized in Table 5. Once again, the full dataset is unbalanced, we then choose $\mathcal{D}_{\mathrm{UK}}$ as a balanced subset of 2,000 patients. This dataset is private.

**Model** We use the same mortality predictor as in the previous experiment (see Table 2). The only difference is that no weight decay is included in the optimization ($\lambda = 0$).

| Feature | Range |
|---|---|
| Age | $64 - 76$ |
| PSA | $8 - 21$ |
| Comorbidities | $0, 1, 2, \geq 3$ |
| Treatment | Hormone Therapy (PHT), Radical Therapy - RDx (RT-RDx), |
| | Radical Therapy -Sx (RT-Sx), CM |
| Grade | $1, 2, 3, 4, 5$ |
| Stage | $1, 2, 3, 4$ |
| Primary Gleason | $1, 2, 3, 4, 5$ |
| Secondary Gleason | $1, 2, 3, 4, 5$ |

Table 5: Features for the CUTRACT Dataset.

**Note on the prostate cancer datasets** In the medical literature on prostate cancer [8], the grade of a patient can be deduced from the Gleason scores in the following way:

$$\text{Gleason1} + \text{Gleason2} \leq 6 \implies \text{Grade} = 1$$
$$\text{Gleason1} = 3 \wedge \text{Gleason2} = 4 \implies \text{Grade} = 2$$
$$\text{Gleason1} = 4 \wedge \text{Gleason2} = 3 \implies \text{Grade} = 3$$
$$\text{Gleason1} + \text{Gleason2} = 8 \implies \text{Grade} = 4$$
$$\text{Gleason1} + \text{Gleason2} \geq 9 \implies \text{Grade} = 5$$

We noted that this relationship between the grade and the Gleason score was not always verified among the patients in our two prostate cancer datasets. After discussing with our curator, we understood that the data of some patients was collected by using a different convention. Further, some of the data was missing and has been imputed in a way that does not respect the above rule. Clearly, those details are irrelevant if we use this data to train a model to illustrate the functionalities of SimplEx as it is done in our paper. Nonetheless, it should be stressed that this inconsistency with the medical literature implies that our models are only illustrative and should not be used in a medical context. To avoid any confusion, we have removed the Gleason scores from the Figures in the main paper. For completeness, we have included the Gleason scores in Figures 11,12,13. As we can observe, not all the patients verify the above rule.

### 2.3  Detection of EMNIST letters

**Description** We propose an analogue of the detection of British patient from Section 3.3 in the image classification setting. We train a CNN with a training set extracted from the MNIST dataset $\mathcal{D}_{\text{train}} \subset \mathcal{D}_{\text{MNIST}}$. Next, we sample a corpus $\mathcal{C} \subset \mathcal{D}_{\text{train}}$ of size $C = 1,000$. We are now interested in investigating if the latent representation of test examples from another similar dataset can be distinguished from latent representations of MNIST examples. To that aim, we use the EMNIST-Letter dataset $\mathcal{D}_{\text{EMNIST}}$, which contains images that are similar to MNIST images. There is one major difference between the two datasets: EMNIST-Letter images represent letters, while MNIST images represent numbers. To evaluate quantitatively if this difference matters for the model representation, we consider a mixed set of test examples $\mathcal{T}$ sampled from both $\mathcal{D}_{\text{MNIST}}$ and $\mathcal{D}_{\text{EMNIST}}$: $\mathcal{T} \subset \mathcal{D}_{\text{MNIST}} \sqcup \mathcal{D}_{\text{EMNIST}}$. We sample 100 examples from both sources: $|\mathcal{T} \cap \mathcal{D}_{\text{MNIST}}| = |\mathcal{T} \cap \mathcal{D}_{\text{EMNIST}}| = 100$. For the rest, we follow the same procedure as in the clinical use-case: we approximate the latent representation of each example $\mathbf{h} \in \mathbf{g}(\mathcal{T})$, compute the associated corpus residual $r_{\mathcal{C}}(\mathbf{h})$ and sort the examples by decreasing order of residual.

**Metrics** We use the same metrics as in Section 2.2.

**Baselines** We use the same baselines as in Section 2.2.

**EMNIST-Letter dataset** EMNIST-Letter contains 145,600 images, each representing a handwritten letter [9]. These images have exactly the same format as MNIST images: $28 \times 28$ pixels with one

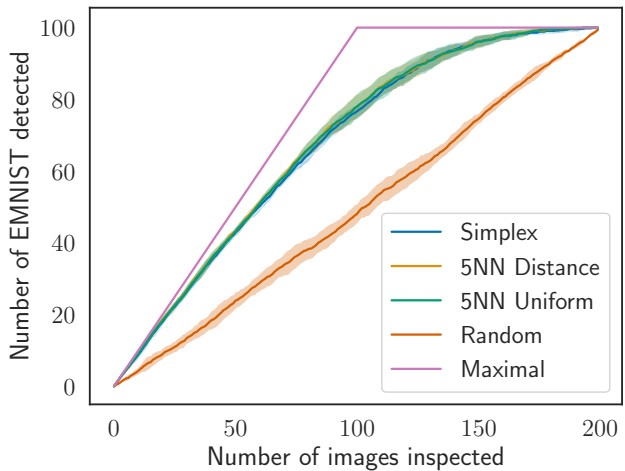

Figure 4: Detection of EMNIST examples.

channel. Ryan Cooper holds the copyright of EMNIST dataset, which is a derivative work from original MNIST datasets. EMNIST dataset is made available under the terms of the MIT license.

**Model** As in the previous experiment, we train the CNN from Table 3. In contrast with the previous experiment, we set the weight decay to zero: $\lambda = 0$. The resulting model has more than $99\%$ accuracy on the test set.

**Results** Results are shown in Figure 4. This suggests that the difference between the two dataset is encoded in their latent representation. SimplEx and the baselines offer similar performances in this case.

### 2.4 Experiments with synthetic time series

In this section, we describe some further experiments we have performed with synthetic time series. The two parts of this subsection mirror the experiments that we have performed in Section 3 of the main paper.

#### 2.4.1 Corpus precision

**Description** This experiment mirrors the experiments from Section 3.1 of the main paper. We start with a time series dataset $\mathcal{D}$ that we split into a training set $\mathcal{D}_{\text{train}}$ and a testing set $\mathcal{D}_{\text{test}}$. We train a black-box $f$ for a time series forecasting task on the training set $\mathcal{D}_{\text{train}}$. We randomly sample a set of corpus examples from the training set $\mathcal{C} \subset \mathcal{D}_{\text{train}}$ (we omit the true labels for the corpus examples) and a set of test examples from the testing set $\mathcal{T} \subset \mathcal{D}_{\text{test}}$. For each test example $\mathbf{x} \in \mathcal{T}$, we build an approximation $\hat{\mathbf{h}}$ for $\mathbf{h} = \mathbf{g}(\mathbf{x})$ with the corpus examples latent representations. In each case, we let the method use only $K$ corpus examples to build the approximation. We repeat the experiment for several values of $K$.

**Metrics** We use the same metrics as in Section 3.1 of the main paper. We run the experiment 5 times to report standard deviations across different runs.

**Baselines** We use the same baselines as in Section 3.1 of the main paper.

**Data generation** We generate data from the following AR(2) generating process.

$$x_t = \varphi_1 \cdot x_{t-1} + \varphi_2 \cdot x_{t-2} + \epsilon_t \quad \forall t \in [3 : T + 1], \tag{2}$$

where $\varphi_1 = .7$, $\varphi_2 = .25$ and $\epsilon_t \sim \mathcal{N}(0, 0.1)$. The initial condition for the time series are sampled independently: $x_1, x_2 \sim \mathcal{N}(0, 1)$. Each instance in the dataset $\mathcal{D}$ consists in a couple $(\mathbf{x}, \mathbf{y}) \in \mathcal{D}$ of sequences $\mathbf{x} = (x_t)_{t=1}^T$ and $\mathbf{y} = (y_t)_{t=1}^T$. For each time step, the target indicates the value of the time series at the next step: $y_t = x_{t+1}$ for all $t \in [1 : T]$. We generate 10,000 such instances that we split into 9,000 training instances $\mathcal{D}_{\text{train}}$ and 1,000 testing instances $\mathcal{D}_{\text{test}}$.

| Layer | Input Dimension | Output Dimension | Activation | Remark |
|-------|-----------------|------------------|------------|--------|
| LSTM 1 | $t \times 1$ | $t \times 100$ | | |
| LSTM 2 | $t \times 100$ | 100 | | Output: $\mathbf{h}_t = \mathbf{g}(\mathbf{x}_{1:t})$ |
| Linear | 100 | 1 | | Output: $y_t = f(\mathbf{x}_{1:t})$ |

Table 6: AR Forecasting LSTM, $t$ denotes the length of the input sequence.

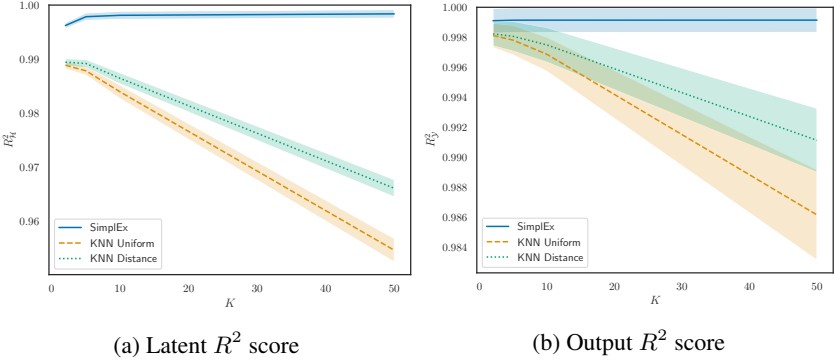

(a) Latent $R^2$ score

(b) Output $R^2$ score

Figure 5: Precision of corpus decomposition for AR (avg $\pm$ std).

**Model** We train a two layer LSTM to forecast the next value of the time series at each time step. The precise model architecture is described in Table 6. The model is trained by minimizing the following loss:

$$\mathcal{L}_{\text{train}}(\boldsymbol{\theta}) = \sum_{(\mathbf{x}, \mathbf{y}) \in \mathcal{D}_{\text{train}}} \sum_{t=1}^{T} \left[ f_{\boldsymbol{\theta}}(\mathbf{x}_{1:t}) - y_t \right]^2,$$

where $\mathbf{x}_{1:t} \equiv (x_t)_{t=1}^{T}$. We train this model with Adam (default Pytorch hyperparameters) for 20 epochs. Across the different runs, the average RMSE of the resulting model on testing data is always 0.1. This corresponds to ideal performances for a deterministic model due to the noise term $\epsilon_t$ in the AR model.

**Results** The results of this experiment are shown in Figure 5 & 6. As in Section 3.1 of the main paper, SimplEx offers significantly better and more consistent results across different runs.

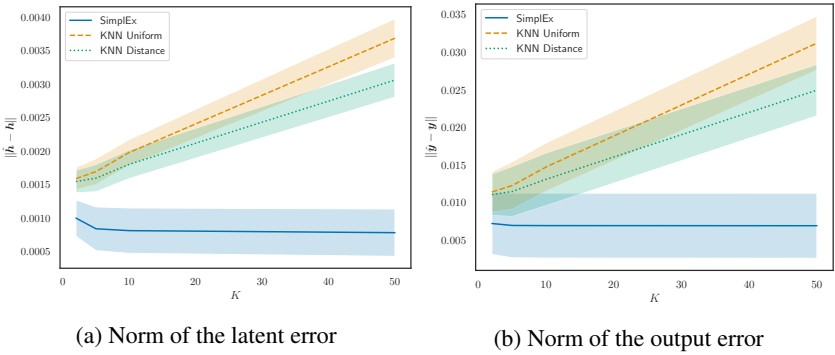

(a) Norm of the latent error

(b) Norm of the output error

Figure 6: Precision of corpus decomposition for AR (avg $\pm$ std).

### 2.4.2 Detection of oscillating time series

**Description** This experiment mirrors the EMNIST detection experiment from Section 2.3. We train a LSTM with a training set extracted from the previous AR dataset $\mathcal{D}_{\text{train}} \subset \mathcal{D}$. Next, we sample a corpus $\mathcal{C} \subset \mathcal{D}_{\text{train}}$ of size $C = 1,000$. We are now interested in investigating if the latent representation of test examples from another similar dataset can be distinguished from latent representations of traditional AR examples. To that aim, we use a dataset sampled from a distinct AR(2) process $\tilde{\mathcal{D}}$. To evaluate quantitatively if this difference matters for the model representation, we consider a mixed set of test examples $\mathcal{T}$ sampled from both $\mathcal{D}$ and $\tilde{\mathcal{D}}$: $\mathcal{T} \subset \mathcal{D}_{\text{test}} \sqcup \tilde{\mathcal{D}}$. We sample 1,000 examples from both sources: $|\mathcal{T} \cap \mathcal{D}_{\text{test}}| = |\mathcal{T} \cap \tilde{\mathcal{D}}| = 1,000$. For the rest, we follow the same procedure as in the clinical use-case: we approximate the latent representation of each example $\mathbf{h} \in \mathbf{g}(\mathcal{T})$, compute the associated corpus residual $r_{\mathcal{C}}(\mathbf{h})$ and sort the examples by decreasing order of residual.

**Metrics** We use the same metrics as in Section 2.2. We run the experiment 5 times to report standard deviations across different runs.

**Baselines** We use the same baselines as in Section 2.2.

**Data generation** We generate $\mathcal{D}$ as in the previous experiment. The time series in $\tilde{\mathcal{D}}$ are sampled from the following AR(2) process:

$$\tilde{x}_t = -\varphi_1 \cdot \tilde{x}_{t-1} + \varphi_2 \cdot \tilde{x}_{t-2} + \epsilon_t \quad \forall t \in [3 : T+1], \tag{3}$$

where $\varphi_1$, $\varphi_2$ and $\epsilon_t$ are defined as in (2). The initial condition for the time series are sampled independently: $\tilde{x}_1, \tilde{x}_2 \sim \mathcal{N}(0,1)$. The only difference between $\mathcal{D}$ and $\tilde{\mathcal{D}}$ lies in the extra minus sign from (3) compared to (2). This gives an extra oscillating behaviour to the time series from $\tilde{\mathcal{D}}$. We generate 1,000 such instances that we use for testing purpose.

**Model** We use the same LSTM as in the previous experiment.

**Results** The results of this experiment are shown in Figure 7. As in Section 3.3 of the main paper, the difference between $\mathcal{D}$ and $\tilde{\mathcal{D}}$ is imprinted in their latent representation. Once again, SimplEx offers the best detection scheme.

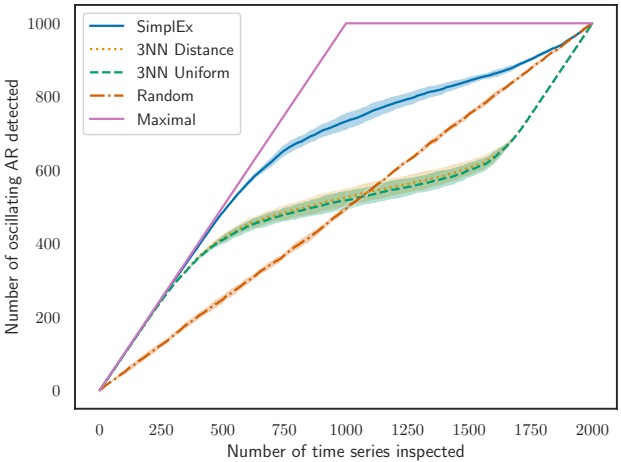

Figure 7: Detection of oscillating AR.

### 2.5 A comparison with Influence Functions

**Description** We note that there is no standard way to reconstruct the explicit black-box output $\mathbf{f}(\mathbf{x})$ with the influence scores [10] for an input $\mathbf{x} \in \mathcal{X} \subset \mathbb{R}^{d_X}$. In contrast, SimplEx allows to explicitly decompose a black-box prediction in terms of contributions arising from each corpus example: $\mathbf{f}(\mathbf{x}) = \sum_{c=1}^{C} w^c \mathbf{l}(\mathbf{h}^c)$. An interesting question to ask is the following: can we interpret influence scores as reconstruction weights in latent space? To explore this question, we propose the following

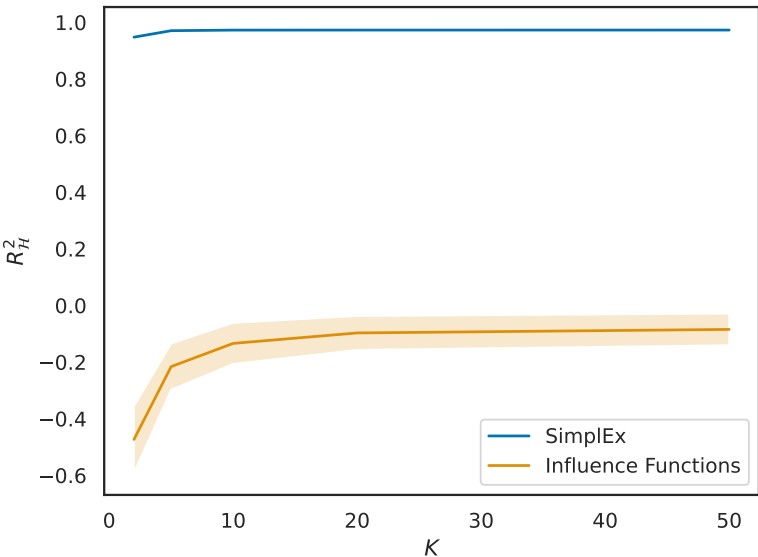

Figure 8: Precision of the corpus decomposition in latent space.

procedure. First, we compute the influence score $i^c \in \mathbb{R}$ for the prediction $\mathbf{f}(\mathbf{x})$ and for each corpus example $\mathbf{x}^c \in \mathcal{C}$. Then, we extract the helpful examples from the corpus: $\mathcal{C}_{help} = \{x^c \in \mathcal{C} \mid i^c > 0\}$. In the same spirit as in Section 3.1 of the main paper, we select the $K$ most helpful examples from $\mathcal{C}_{help}$. We denote their corpus indices as $\mathcal{I} = \{c_1, c_2, \dots, c_K\} \subset [C]$. Finally, we make a corpus decomposition with weights proportional to the influence score:

$$w^c = \begin{cases} \frac{i^c}{\sum_{k \in \mathcal{I}} i^k} & \text{if } c \in \mathcal{I} \\ 0 & \text{else} \end{cases} \qquad (4)$$

**Metrics** We study the quality of influence-based corpus decomposition $\sum_{c=1}^{C} w^c \mathbf{h}^c$ as an approximation of the test example's latent representation $\mathbf{h} = \mathbf{g}(\mathbf{x})$. Therefore, we use the same metrics as in Section 3.1 of the main paper.

**Baseline** We consider SimplEx as a baseline.

**Dataset** We perform the experiment with the MNIST dataset.

**Results** We report the result of this experiment in Figure 8 (average +/- standard deviation over 5 runs). This confirms that influence functions scores are not suitable to decompose the latent representations in terms of the corpus.

## 2.6 More examples

In Figures 9-12, we provide further examples of corpus decompositions with MNIST and SEER.

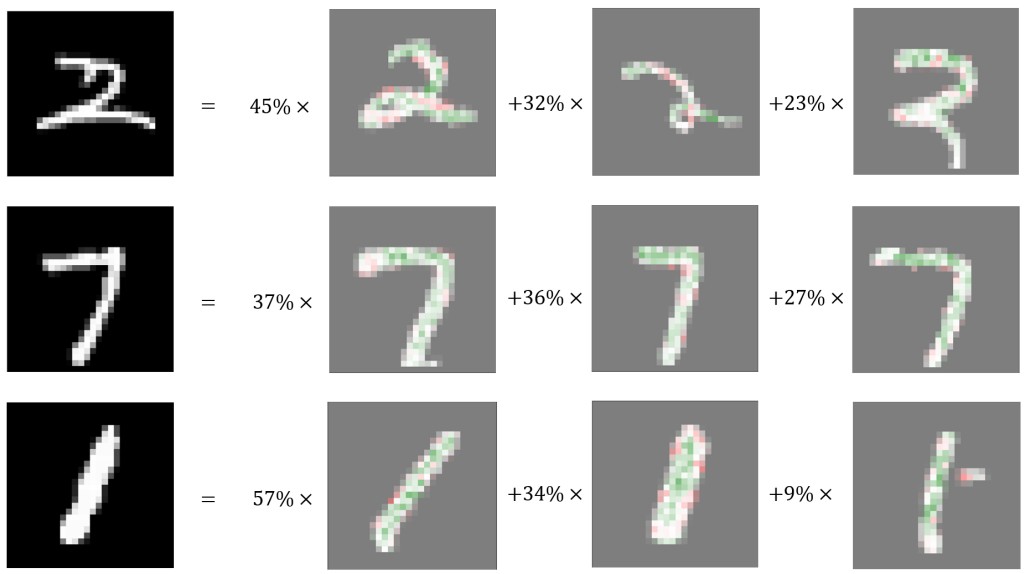

Figure 9: Examples of MNIST decompositions (left: test example, right: corpus decomposition).

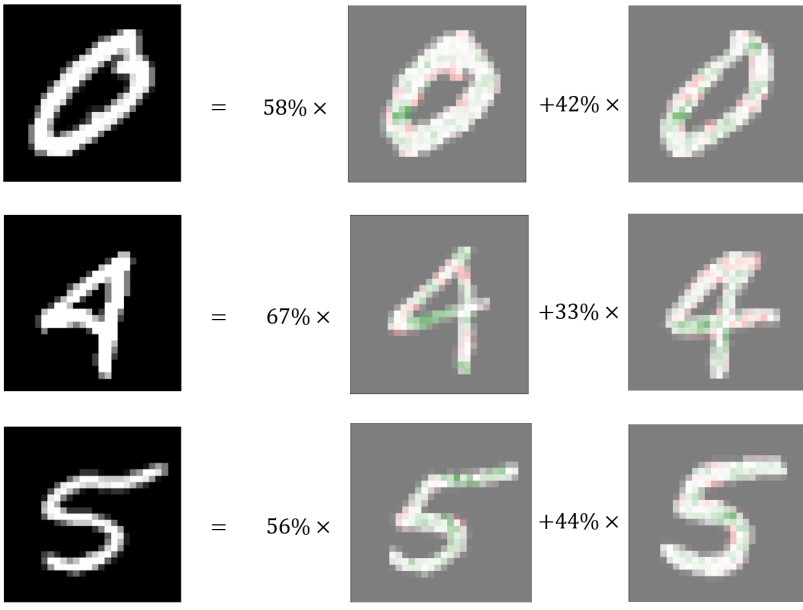

Figure 10: Examples of MNIST decompositions (left: test example, right: corpus decomposition).

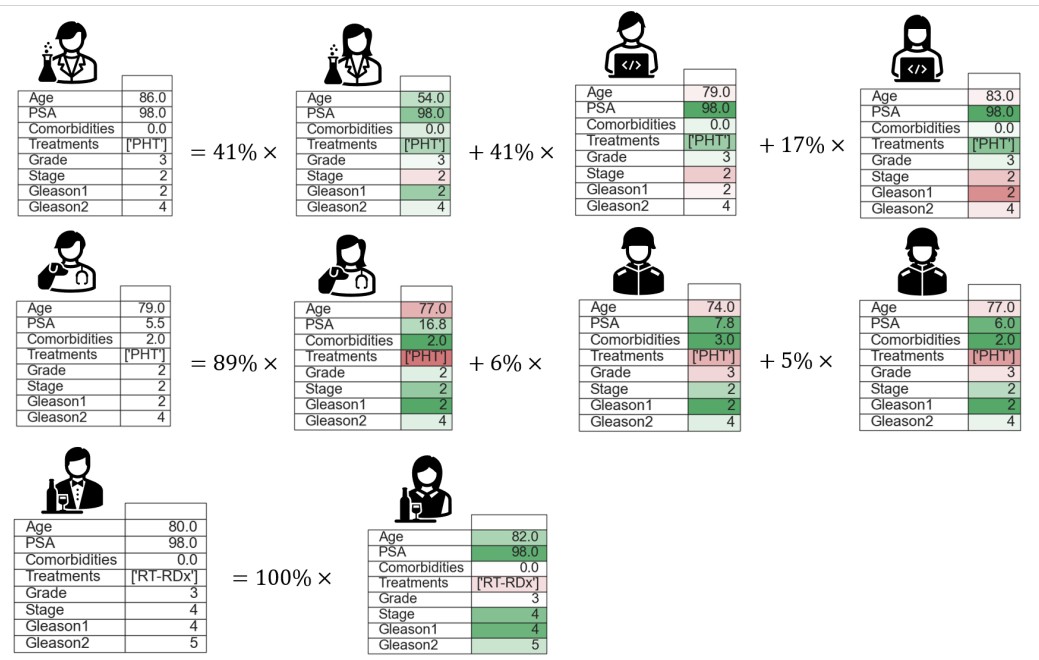

Figure 11: Examples of SEER decompositions (left: test example, right: corpus decomposition).

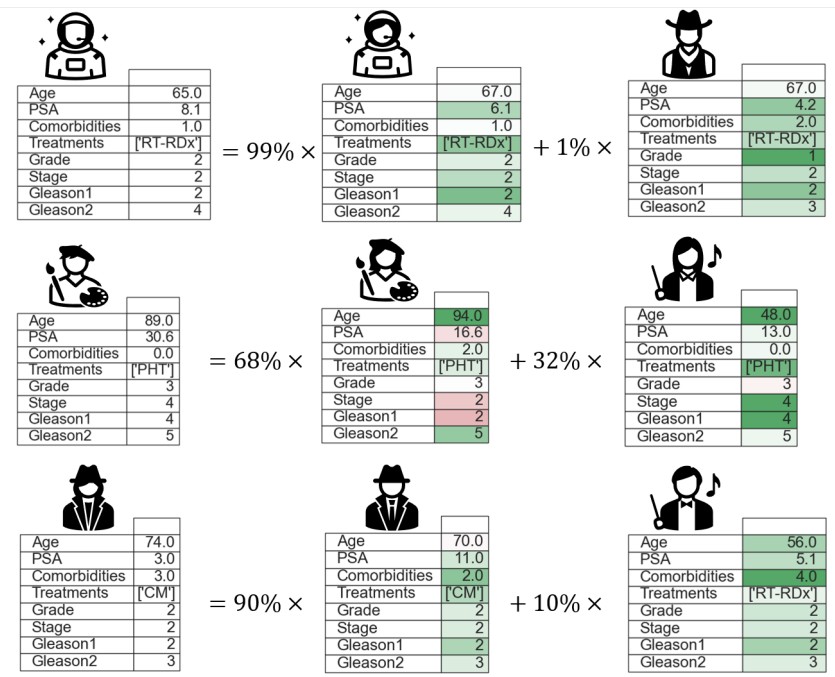

Figure 12: Examples of SEER decompositions (left: test example, right: corpus decomposition).

# 3 User Study

We have conducted a small scale user study with SimplEx. The purpose of this study was to identify if the functionalities introduced by SimplEx are interesting for the clinicians. In total, 10 clinicians took part in the study.

Let us now describe the study. With the SEER Prostate Cancer dataset that is described in the paper, we have performed a SimplEx corpus decomposition presented in Figure 13. The decomposition involved 1 test patient that we called Joe and 2 corpus patients that we called Bill and Max. Our classification model predicted that Joe will die of his prostate cancer. Bill died of his prostate cancer and Max survived. The SimplEx corpus weights were as follows: 66% for Bill, 34% for Max. For both Bill and Max, the Jacobian Projections were given and presented as a measure of importance for each of their features in order to relate them to Joe.

After presenting this explanation to the clinician, we gradually brought their attention to its various components. We made several statements related to SimplEx's functionalities and asked the clinicians if they agree/disagree on a scale from 0 to 5, where 0 corresponds to strongly disagreeing, 3 corresponds to a neutral opinion and 5 corresponds to strongly agreeing.

The first two statements were related to the weights appearing in the corpus decomposition. The purpose was to determine if those are important for the clinicians and if there is an additional value in learning these weights, as is done in SimplEx. The first statement was the following: "The value of the weights in the corpus decomposition is important". The results were the following: 6 of the clinicians agreed (1 strongly), 1 remained neutral and 3 disagreed (1 strongly). The second statement was the following: "Some valuable information is lost in setting the weights to a uniform value" (the doctors are given the KNN Uniform equivalent of SimplEx's explanation in the presented case). The results were the following: 5 of the clinicians agreed (3 strongly), 3 remained neutral, 2 strongly disagreed. We conclude that the majority of the clinicians found the weights to be important. Most of them found that hard-coding the weights as in the KNN Uniform baseline hides some valuable information.

The third statement was related to the Jacobian Projections. The purpose was to determine if the Jacobian Projections provide valuable information for interpretability. The statement was the following: "Knowing which feature increases the similarity/discrepancy between two patients is important". The results were the following: 9 of the clinicians agreed (5 strongly), 1 disagreed. We conclude that the Jacobian Projections constitute a crucial part of SimplEx's explanations.

The fourth statement was related to the freedom of choosing the corpus. The purpose was to determine if the flexibility of SimplEx is useful in practice. The statement was the following: "It is important

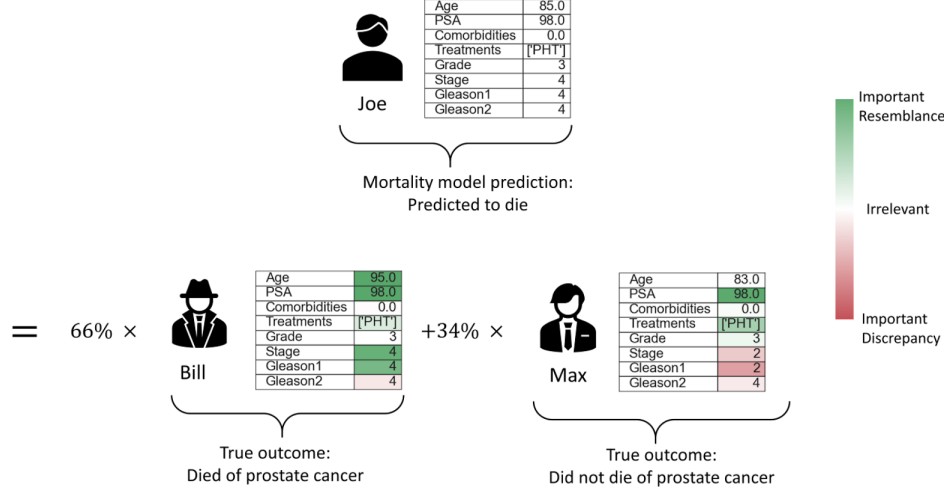

Figure 13: SimplEx example provided in the user study.

for the clinician to be able to choose the patients in the corpus that is used for comparison". The results were the following: 4 of the clinicians agreed (1 strongly), 1 remained neutral, 5 disagreed (3 strongly). Clearly, the clinicians are more divided on this point. However, this additional freedom offered by SimplEx comes at no cost. A clinician that desires explanations in terms of patients they are familiar with can use their own corpus. A clinician that is happy with explanations in terms of any patients can use a corpus sampled from training data.

The last statement was related to the use of SimplEx in order to anticipate misclassification, as it is suggested in Section 3.2 of the main paper. The statement was the following: "If Bill had not died due to his prostate cancer, this would cast doubt on the mortality predicted for Joe". The results were the following: 6 of the clinicians agreed (2 strongly), 1 remained neutral, 3 disagreed (2 strongly). We conclude that, for the majority of the clinicians, SimplEx's explanations affect their confidence in the model's prediction.