# OpenReview forum: "Explaining Latent Representations with a Corpus of Examples"
_NeurIPS.cc/2021/Conference — NeurIPS 2021 Spotlight_

### Official Review · Reviewer_coXq · 2021-07-05

**Rating:** 8
**Confidence:** 4

**Summary:**

This paper introduces SimplEx, which is a method to post-hoc explain a model prediction by computing from a user-defined set of examples, how the set of examples decompose the prediction. For example, the clinical risk prediction of a patient can be explained by a set of other patients and how much each patient influenced the prediction.

Contributions
-	A post-hoc explanation method, called SimplEx, were users can choose the set of examples from which the explanation should be constructed. This is advantageous when the training set is not available or a user wants customized explanations, e.g. a doctor wanting to understand a predictions in terms of other patients she/he knows.
-	SimplEx combines the ideas of feature saliency and example-based explanations. For each example, the method computes the percentage of this example contributing to explaining the prediction as well as which features of this example were important.
-	A latent-space generalization of Integrated Gradients [10]
-	Experimental evaluation that show SimplEx to be more precise and robust compared to baselines.


**Limitations And Societal Impact:**

A broader impact section could be added. In a separate section (e.g. supplementary material), there could be an explicit discussion on when the method should not be used, e.g. as shown in Figure 8, the American corpus shouldn’t be used to explain the British patient. Also see last question above – what if we don’t know that the patient is British? Can this be detected? This should also be discussed in such a section.

**Main Review:**

Strengths
-	The paper is well written and easy to follow
-	The method is interesting and supported by experimental results

Weaknesses
-	The paper leaves some natural questions open (see questions below).
-	Line 170 mentions that the corpus residual can be used to detect an unsuitable corpus, but there are no experiments to support this.

After authors' response
All the weakness points have been addressed by the authors' response. Consequently I have raised my score. In particular:
- The left open questions have all been answered.
- There indeed is an experiment to support this, thanks to the authors' for clarifying this, that connection was not clear to me previously.

Questions
-	Line 60: Why do you say that e.g. influence functions cannot be used to explain a prediction? The explanation of a prediction could be the training examples whose removal (as determined by the influence function) would lead to the largest score drop for a prediction.
-	How does the method scale as the corpus size or hidden dimension size is increased?
-	What happens if a too small corpus is chosen? Can this be detected?
-	What if we don’t know that a test example is crucially different, e.g. what if we don’t know that the patient of Figure 8 is “British” and we use the American corpus to explain it? Can this be detected with the corpus residual value?
-	In the supplementary material you mention how it is possible to check if a decomposition is unique. Do you do this in practice when conducting experiments? How do you choose a decomposition if it is not unique? What does it imply for the experiments (and the usage of the method in real-world applications) if the decomposition is not unique?

Typos, representation etc.
-	Line 50: An example of when a prototype model would be unsuitable would strengthen your argument.
-	Footnote 2: “or” -> “of”
-	Line 191: when the baseline is first introduced, [10] or other references would be helpful to support this approach
-	Line 319: “the the” -> “the”
-	Line 380: “at” -> “to”?


**Time Spent Reviewing:**

3.5

---

> ### Author Response · Authors · 2021-08-09
> **Response to Reviewer coXq [Part 2/2]**
>
> ### (5) Uniqueness of the corpus decomposition
>
> Most corpora used in our experiment are too large to guarantee the uniqueness of the decomposition. Indeed, we typically choose $C \gg d_H$ so that the uniqueness of the corpus decomposition cannot be guaranteed by Proposition 1.2 from the supplementary material.
>
> If one wants to guarantee the uniqueness of the corpus decomposition, there is an interesting trade-off to take into account. Clearly, from Proposition 1.2 of the supplementary material, guaranteeing the uniqueness limits the size of the corpus. As explained in Section 1.2 of the supplementary material, this stems from the fact that large corpora typically contain examples that produce redundant latent representations. However, as we have previously discussed, larger corpora have a bigger approximation power in latent space.
>
> Since the weights are directly optimized in Algorithm 1 from the supplementary material, the optimization automatically converges to one such optimal decomposition so this choice is not made by us. We don't regard this as a problem in practice as all of these decompositions are equally good and their multiplicity is just an artifact of the  corpus redundancy.
>
> ---
>
>
> ### (6) Suggested revisions
> Many thanks for the suggestions, we will correct the typos in the revised version.
>
> ---

---

> ### Author Response · Authors · 2021-08-09
> **Response to Reviewer coXq [Part 1/2]**
>
> Thank you for your thoughtful comments and suggestions. We give answers to each in turn.
>
> ---
>
> ### (1) Detect unsuitable corpus and examples with the residual
>
> Many thanks for the interesting questions on using the residuals to detect unsuitable corpora/ test examples. We believe that they are closely related so we will give a unified answer. The corpus residual indeed allows to detect examples for which the corpus decomposition performed by SimplEx cannot be trusted. Checking this is precisely the purpose of  the experiment presented in Figure 6 of the main paper and in Figure 4, 7 of the supplementary materials.
>
> More precisely, the purpose of these outlier detection experiments is not to demonstrate that SimplEx is the best outlier detection scheme but to demonstrate that outliers will typically be given a latent representation that is more likely to be poorly approximated by the corpus and that SimplEx's corpus residual makes it possible to highlight such examples. This information is important since it measures how accurate the SimplEx decomposition is (and hence measures to what extent SimplEx's explanation can be trusted).
>
> It is interesting to make the whole setup more formal to better understand the significance of this experiment. We suppose that our black-box $\textbf{f}$ has been trained with a training set $\mathcal{D}^{train}$ sampled from a data generating process $\mathcal{P}$. In Section 3 of the main paper and Section 2 of the supplementary, we study 3 such data generating processes: MNIST, American Prostate Cancer Patients and AR(2) Time Series. We want to use SimplEx with a corpus $\mathcal{C} \subset \mathcal{D}^{train}$ sampled from the same data generating process  $\mathcal{P}$.
>
> Now assume that we have another data generating process $\mathcal{P}^{out}$ that is distinct from $\mathcal{P}$.  In our paper, we consider 3 such alternative data generating processes: EMNIST (images of letters), British Prostate Cancer Patients and Oscillating AR(2) Time Series. It might be inappropriate to interpret examples sampled from $\mathcal{P}^{out}$ with our corpus $\mathcal{C}$ since they arise from distinct data generating processes. If this is indeed the case, it is useful to have a metric to evaluate the extent to which the explanations can be trusted. The corpus residual $r_{\mathcal{C}}(\textbf{h})$ precisely plays this role: when it is large, we know that the test sample is poorly approximated by the corpus *in latent space*. Conversely, when it is small, we know that the test sample is well approximated by the corpus *in latent space*.
>
> The experiments associated to Figure 6 of the main paper and Figures 4, 7 of the supplementary material demonstrate that test examples sampled from $\mathcal{P}^{out}$ indeed have a typically larger corpus residuals that test examples sampled from $\mathcal{P}$. This is reflected by the fact that sorting the examples of a mixed set (sampled from both $\mathcal{P}^{out}$ and  $\mathcal{P}$) by decreasing order of corpus residual allows to detect examples that were sampled from  $\mathcal{P}^{out}$ significantly better than at random. Since these examples typically have a larger corpus residual, the user will rightfully be more sceptical toward the explanations by SimplEx in these cases.
>
> ---
>
> ### (2) Influence Functions and Explainability
>
> Thank you for this remark, it is true that the related work discussion with respect to influence functions could be better formulated. What we meant is that there is no standard way to reconstruct the explicit black-box output $\textbf{f}(\textbf{x})$ with the influence scores for an input $\textbf{x} \in \mathcal{X} \subset \mathbb{R}^{d_X}$. In contrast, SimplEx allows to explicitly decompose a black-box prediction in terms of contributions arising from each corpus example: $\textbf{f}(\textbf{x}) = \sum_{c=1}^C w^c \textbf{l}(\textbf{h}^c)$.
>
> An interesting question to ask is the following: can we interpret influence scores as reconstruction weights in latent space? To explore this question, we propose the following procedure:
>
> 1. Compute the influence score $i^c \in \mathbb{R}$ for the prediction $\textbf{f}(\textbf{x})$ and for each corpus example $\textbf{x}^c \in \mathcal{C}$.
> 2. Extract the helpful examples from the corpus:  $\mathcal{C}_{help} = \\{ x^c \in \mathcal{C} \mid i^c > 0 \\}$.
> 3. In the same spirit as in Section 3.1 of the main paper, select the $K$ most helpful examples from  $\mathcal{C}_{help}$. We denote their corpus indices as $\mathcal{A} = \\{ c_1, c_2, \dots, c_K \\} \subset [C]$.
> 4. We make a corpus decomposition with weights proportional to the influence score:
>      $$w^c = I_{\mathcal{A}}(c)   \frac{i^c}{\sum_{a \in \mathcal{A}}i^{a}},$$
>      where $I_{\mathcal{A}}$ denotes the indicator function on $\mathcal{A}$.
> 5. We study the quality of influence-based corpus decomposition $\sum_{c=1}^C w^c \textbf{h}^c$ as an approximation of the test example's latent representation $\textbf{h} = \textbf{g}(\textbf{x})$.
>
> We perform this experiment in precisely the same setup as the precision experiment for MNIST in Section 3.1 of the main paper.  We report the results of this experiment in the below Table (average +/- standard deviation over 5 runs):
>
> |  Method |  $R^2_{\mathcal{H}} (K=2)$ 	|  $R^2_{\mathcal{H}} (K=5)$ | $R^2_{\mathcal{H}} (K=10)$ |   $R^2_{\mathcal{H}} (K=20)$ |  $R^2_{\mathcal{H}} (K=50)$ |
> |---				   |	---                                            |				---								 | --- | --- | --- |
> |   	SimplEx |   0.96 +/- 0.00 | 0.98 +/- 0.00 | 0.98 +/- 0.00 | 0.98 +/- 0.00 | 0.98 +/- 0.00 |
> |   Infl. Funct.	|   -0.53 +/- 0.05 | -0.21 +/- 0.03 | -0.12 +/- 0.03 | -0.07 +/- 0.01 | - 0.03 +/- 0.01
>
> This confirms that influence functions scores are not suitable to decompose the latent representations in terms of the corpus.
>
> ---
>
> ### (3) Scaling of SimplEx
>
> To answer this question, it is instructive to do a complexity analysis of Algorithm 1 and 2 from the supplementary material. Let $C$ be the corpus size, $d_X$ and  $d_H$ be the dimensions of the input space and the latent space respectively. SimplEx requires to precompute the latent representation of each corpus example. Let us denote by $L$ the price of evaluating the latent representation of an example (this clearly depends on the number of parameters in the black-box model). Therefore, the initialization of SimplEx requires $\mathcal{O}( L\cdot C)$ computations.
>
> Let us start by Algorithm 1. We assume that the optimization requires $N_o$ loops. Computing the softmax required to normalize the weights corresponds to $\mathcal{O}(C)$ operations. Evaluating the loss requires $\mathcal{O}(C+d_H)$ operations. Finally, performing the Adam step requires $\mathcal{O}(C)$ operations. By putting everything together, the complexity of Algorithm 1 is $\mathcal{O}(N_o(C+d_H) + L\cdot C)$.
>
> Now let us perform the same analysis for Algorithm 2. Forming the input and baseline matrices requires $\mathcal{O}(C \cdot d_X)$.  Now let us denote by $N_b$ the number of bins involved in the approximation of the Jacobian Projection. Further, we denote by $J$ the cost of evaluating a Jacobian-vector product for the the map $\textbf{g}$ between the input and the latent space (this clearly depends on the number of parameters in the black-box model). Clearly, the bottleneck for each bin is to perform the increment to the Jacobian projection. This task has a complexity  $\mathcal{O}(J \cdot C \cdot d_X)$. By putting everything together, , the complexity of Algorithm 2 is $\mathcal{O}(C (J \cdot d_X \cdot N_b+L))$.
>
> Hence, the complexity of computing the corpus weights is $\mathcal{O}(N_o(C+d_H) + L\cdot C)$ and the complexity of computing the Jacobian Projections is $\mathcal{O}(C (J \cdot d_X \cdot N_b+L))$. Performing the whole SimplEx analysis hence scales as $\mathcal{O}(C (J \cdot d_X \cdot N_b+ N_o + L) + N_o \cdot d_H)$. In conclusion, SimplEx scales linearly with the size of the corpus and the dimension of the latent space. That being said, we believe that the bottleneck lies rather in the evaluation of the Integrated Jacobians, which is optional if one is not interested in an explanation at the feature level (i.e. one is happy with a weighted corpus decomposition in latent space).
>
>
> ---
>
> ### (4) Size of the corpus
>
> It is indeed legitimate to ask what happens when smaller corpora are chosen. In reducing the size of the corpus, one should expect the corpus residual $r_{\mathcal{C}}(\textbf{h})$ to decrease as the approximation power of a corpus will typically decrease with its size. To make this more quantitative, we consider the setting of Section 3.1 from the main paper and we report the average corpus residual as a function of the corpus size in the below Table (mean +/- standard deviation across 5 runs):
>
> |  Corpus size |  $r_{\mathcal{C}}(\textbf{h})$  |
> | --- |	--- |
> |  50 |   0.32 +/- 0.03 |
> |  100 | 0.26 +/- 0.03  	|
> |   500	|  0.16 +/- 0.03 	|
> |   1000	|  0.12 +/- 0.01 	|
>
> We clearly see that larger corpora allow to have better approximations in latent space (i.e. smaller residuals). Note that, as suggested by Section 3.1 of the main paper, even corpora of size 100 correspond to good approximations (high $R^2$ score) in this case. In practice, a low $R^2$ score and a large corpus residual hint that a corpus is too small or unsuitable to explain the selected examples.
>
> ---

---

### Official Review · Reviewer_QnMz · 2021-07-13

**Rating:** 7
**Confidence:** 3

**Summary:**

This paper introduces SimplEx, a method that approximate the hidden representation of a test example using the linear combination of hidden representations of examples in a pre-defined corpus. Further, the hidden representation of the test instance is associated with each input feature of these examples in the corpus. The paper demonstrates the quality of approximation of SimplEx (Sec 3.1, SimplEx outperforms kNN baselines) and demonstrate its practical utility in the experiment of predicting cancer mortality (Sec 3.2) by allowing model users to interpret model predictions and determine whether to trust the prediction. Detailed math derivation and additional experiments with MNIST and time series data are included in the appendix.

**Limitations And Societal Impact:**

The authors describes assumption 2.1 as the main limitation to their approach. My concerns and questions are placed in the "main review" section.
I believe the work does not have foreseeable negative societal impact.

**Main Review:**

Strength:
- A novel framework to interpret neural classification models with minimal assumptions on the model architecture.
- SimplEx bridges attribution-based (feature-based) interpretation methods and example-based interpretation method.
- The intuition of SimplEx is reasonable and mimics human learning process (by retrieving and comparing to past examples).
- Writing is coherent and self-contained. Illustrations are clear.

Weakness:
- It is questionable whether introducing weights (w^c) assigned to examples from the corpus improves interpretability empirically. The work may benefit from additional human evaluation, e.g., showing interpretation produced by SimplEx (Fig. 7) and produced by kNN Uniform/Distance/Random (hard-coding w_c).
- The SimplEx method may be computationally expensive (precompute the projected Jacobian for every instance in the corpus, perform a convex optimization procedure for every test instance). Some discussion on this topic would be helpful.

Questions:
- The assumptions in the Sec 3.2 use case is very strong — The doctors are required to withhold 1000 labeled examples to use SimplEx. Is this a hard constraint? How does the explanation quality change when fewer datapoints are used.
- I find it hard to understand Fig 2 and the related experiment. Does the figure suggest the patients with top 100 largest residues are supposed to be UK patients in the ideal setting (as their representations should be different from those of USA patients)? Does this suggest SimplEx can also be used for outlier/anomaly detection?

**Time Spent Reviewing:**

4

---

> ### Author Response · Authors · 2021-08-09
> **Response to Reviewer QnMz [Part 2/2]**
>
> ### (4) Corpus residual and outlier detection
>
> Thank you for pointing this out, we believe that our name *ideal* baseline is indeed misleading as it would suggest that we are trying to design an optimal outlier detection scheme. This baseline should probably be named *maximal* as it corresponds to the maximal number of outliers that could be detected with a given number of inspections (the same goes for Figure 4 and 7 of the supplementary material). We will rename it accordingly in the revised manuscript.
>
> The purpose of these outlier detection experiments is not to demonstrate that SimplEx is the best outlier detection scheme but to demonstrate that outliers will typically be given a latent representation that is more likely to be poorly approximated by the corpus and that SimplEx's corpus residual makes it possible to highlight such examples. This information is important since it measures how accurate the SimplEx decomposition is (and hence measures to what extent SimplEx's explanation can be trusted).
>
> It is interesting to make the whole setup more formal to better understand the significance of this experiment. We suppose that our black-box $\textbf{f}$ has been trained with a training set $\mathcal{D}^{train}$ sampled from a data generating process $\mathcal{P}$. In Section 3 of the main paper and Section 2 of the supplementary, we study 3 such data generating processes: MNIST, American Prostate Cancer Patients and AR(2) Time Series. We want to use SimplEx with a corpus $\mathcal{C} \subset \mathcal{D}^{train}$ sampled from the same data generating process  $\mathcal{P}$.
>
> Now assume that we have another data generating process $\mathcal{P}^{out}$ that is distinct from $\mathcal{P}$.  In our paper, we consider 3 such alternative data generating processes: EMNIST (images of letters), British Prostate Cancer Patients and Oscillating AR(2) Time Series. It might be inappropriate to interpret examples sampled from $\mathcal{P}^{out}$ with our corpus $\mathcal{C}$ since they arise from distinct data generating processes. If this is indeed the case, it is useful to have a metric to evaluate the extent to which the explanations can be trusted. The corpus residual $r_{\mathcal{C}}(\textbf{h})$ precisely plays this role: when it is large, we know that the test sample is poorly approximated by the corpus *in latent space*. Conversely, when it is small, we know that the test sample is well approximated by the corpus *in latent space*.
>
> The experiments associated to Figure 6 of the main paper and Figures 4, 7 of the supplementary material demonstrate that test examples sampled from $\mathcal{P}^{out}$ indeed have a typically larger corpus residuals that test examples sampled from $\mathcal{P}$. This is reflected by the fact that sorting the examples of a mixed set (sampled from both $\mathcal{P}^{out}$ and  $\mathcal{P}$) by decreasing order of corpus residual allows to detect examples that were sampled from  $\mathcal{P}^{out}$ significantly better than at random. Since these examples typically have a larger corpus residual, the user will rightfully be more sceptical toward the explanations by SimplEx in these cases.
>
> ---

---

> > ### Comment · Reviewer_QnMz · 2021-08-23
> > **Reply**
> >
> > Thank you for your detailed response. I am convinced that (1) weights assigned to the examples aligned with human judgments, according to the user study. (2) The computation cost of SimplEx is acceptable. (3) SimplEx approximates well even with a small corpus. I will increase my score.

---

> > > ### Author Response · Authors · 2021-08-30
> > > **Reply**
> > >
> > > Many thanks for this feedback and for this thorough review. We will improve the manuscript with these useful comments.

---

> ### Author Response · Authors · 2021-08-09
> **Response to Reviewer QnMz [Part 1/2]**
>
> Thank you for your thoughtful comments and suggestions. We give answers to each in turn.
>
> ---
>
> ### (1) User study
>
>
> As suggested, we have conducted a small scale user study with SimplEx. The purpose of this study was to identify if the functionalities introduced by SimplEx are interesting for the clinicians. In total, 10 clinicians took part in the study.
>
> Let us now describe the study. With the SEER Prostate Cancer dataset that is described in the paper, we have performed a SimplEx corpus decomposition such as the one presented in Figure 1 of the main paper. The decomposition involved 1 test patient that we called Joe and 2 corpus patients that we called Bill and Max. Our classification model predicted that Joe will die of his prostate cancer. Bill died of his prostate cancer and Max survived. The SimplEx corpus weights were as follows: 66% for Bill, 34% for Max. For both Bill and Max, the Jacobian Projections were given and presented as a measure of importance for each of their features in order to relate them to Joe.
>
> After presenting this explanation to the clinician, we gradually brought their attention to its various components. We made several statements related to SimplEx's functionalities and asked the clinicians if they agree/disagree on a scale from 0 to 5, where 0 corresponds to strongly disagreeing, 3 corresponds to a neutral opinion and 5 corresponds to strongly agreeing.
>
> The first two statements were related to the weights appearing in the corpus decomposition. The purpose was to determine if those are important for the clinicians and if there is an additional value in learning these weights, as it is done in SimplEx. The first statement was the following: “The value of the weights in the corpus decomposition is important”. The results were the following: 6 of the clinicians agreed (1 strongly), 1 remained neutral and 3 disagreed (1 strongly). The second statement was the following: “Some valuable information is lost in setting the weights to a uniform value” (the doctors are given the KNN Uniform equivalent of SimplEx's explanation in the presented case). The results were the following:  5 of the clinicians agreed (3 strongly), 3 remained neutral, 2 strongly disagreed. We conclude that the majority of the clinicians found the weights to be important. Most of them found that hard-coding the weights as in the KNN Uniform baseline hides some valuable information.
>
> The third statement was related to  the Jacobian Projections. The purpose was to determine if the Jacobian Projections provide valuable information for interpretability. The statement was the following: “Knowing which feature increases the similarity/discrepancy between two patients is important”. The results were the following: 9 of the clinicians agreed (5 strongly), 1 disagreed. We conclude that the Jacobian Projections constitute a crucial part of SimplEx's explanations.
>
> The fourth statement was related to the freedom of choosing the corpus. The purpose was to determine if the flexibility of SimplEx is useful in practice. The  statement was the following: “It is important for the clinician to be able to choose the patients in the corpus that is used for comparison”. The results were the following: 4 of the clinicians agreed (1 strongly), 1 remained neutral, 5 disagreed (3 strongly). Clearly, the clinicians are more divided on this point. However, this additional freedom offered by SimplEx comes at no cost. A clinician that desires explanations in terms of patients they are familiar with can use their own copus. A clinician that is happy with explanations in terms of any patients can use a corpus sampled from training data.
>
> The last statement was related to the use of SimplEx in order to anticipate misclassification, as it is suggested in Section 3.2 of the main paper. The  statement was the following:  “If Bill had not died due to his prostate cancer, this would cast doubt on the mortality predicted for Joe”. The results were the following: 6 of the clinicians agreed (2 strongly), 1 remained neutral, 3 disagreed (2 strongly). We conclude that, for the majority of the clinicians, SimplEx's explanations affect their confidence in the model's prediction.
>
> ---
> ### (2) Computation cost of SimplEx
>
> Let us start with a comparison between SimplEx and our KNN baseline. We believe that a fair comparison between SimplEx and KNN in terms of speed should exclude the evaluation of the Integrated Jacobians. Indeed, SimplEx uses this step to attribute an importance score to each feature of each example from the corpus. This functionality is absent from the pure KNN approach that only involves finding the K-Nearest corpus neighbours in latent space. Let us now give a more concrete discussion of the comparable method. Once again, we want to explain the black-box prediction $\textbf{f}(\textbf{x})$ for an input $\textbf{x} \in \mathcal{X} \subset \mathbb{R}^{d_X}$.
>
> A KNN approach involves $C$ calls of the black-box model (we need to evaluate the latent representation $\textbf{h}^c$ of each corpus example $\textbf{x}^c$ for $c \in [C]$). Then, the KNN ranks the corpus examples by decreasing distance with respect to the latent representation $\textbf{h}$ of $\textbf{x}$ (i.e. decreasing $\vert\vert \textbf{h} - \textbf{h}^c \vert\vert$) to select the K-Nearest Neighbours. Then, the weights $w^c$ are deduced without any additional computation.
>
> The part of SimplEx that performs a similar analysis (i.e. computing the weights) corresponds to Algorithm 1 of the Supplementary Material. Concretely, this method also requires $C$ calls of the black-box to evaluate the latent representation $\textbf{h}^c$ for each $c \in [C]$. Once this is done, SimplEx finds the optimal weights $w^c$ without making any additional call to the black-box. Solving the optimization problem corresponding to Equation (1) from the main paper is more expensive than ranking the corpus example as it is done for KNN. However, it corresponds to solving a trivial convex optimization problem that does not require any further call to the black-box.
>
> On the below table, we have put the time taken to evaluate the weights $w^c$ for one test sample and a corpus of size $C=1000$ with our MNIST classifier model described in Table 3 of the supplementary material:
>
>
> |   	Method |  Computation time per test sample (seconds)  	|
> |---				|	---    |
> |   	SimplEx (corpus decomposition)	 |   $(8.1  \pm 0.1 ) \cdot 10^{-3}$ 	|
> |  KNN |  $(3.1  \pm 0.2 ) \cdot 10^{-5}$ 	|
>
> We observe that, indeed, learning the weights costs extra computation time. However, we believe that the gain in precision reported in Section 3.1 of the main paper and Section 2.4.1 of the Supplementary Material clearly justifies that extra cost. Moreover, we believe that both approaches are fast enough for most practical purposes.
>
>
> The evaluation of the Jacobian Projections is more expensive and took $(5.6  \pm 0.0 ) \cdot 10^{-1}$ seconds per sample in the previous setting. Even though this computation time is higher, it is still reasonable when compared to other methods such as influence functions. In the same setting, it took $1.8  \pm 0.0$ seconds per sample to compute the associated corpus influence functions. This stems from the fact that our method does not require the estimation of any Hessian/Hessian-vector product of the model with respect to its parameter.
>
> ---
>
> ### (3) Size of the corpus as a hard constraint
>
> We would like to emphasize that the examples on which SimplEx relies do not need to be withheld: these examples can perfectly be examples sampled from the black-box training set, as in Section 3.1 of the main paper.
>
> It is indeed legitimate to ask if the size of the corpus is a hard constraint. Reducing the size of the corpus is perfectly possible. In doing so, one should expect the corpus residual $r_{\mathcal{C}}(\textbf{h})$ to decrease as the approximation power of a corpus will typically decrease with its size. To make this more quantitative, we consider the setting of Section 3.1 from the main paper and we report the average corpus residual as a function of the corpus size in the below Table (mean +/- standard deviation across 5 runs):
>
> |  Corpus size |  $r_{\mathcal{C}}(\textbf{h})$  |
> | --- |	--- |
> |   50 |   0.32 +/- 0.03	|
> |   100 | 0.26 +/- 0.03  	|
> |   500 |  0.16 +/- 0.03 	|
> |   1000 |  0.12 +/- 0.01 	|
>
> We clearly see that larger corpora allow to have better approximations in latent space (i.e. smaller residuals).
> Note that, as suggested by Section 3.1 of the main paper, even corpora of size 100 correspond to good approximations (high $R^2$ score) in this case.
>
> ---

---

### Official Review · Reviewer_Zgwu · 2021-07-16

**Rating:** 7
**Confidence:** 3

**Summary:**

In this paper, the authors propose to explain the behavior of black-box neural networks by approximating a top-level hidden layer (no non-linearities afterwards, except possibly softmax) as a linear combination of the hidden layers obtained from different examples. These examples may come from the training data, but it is not required. The contribution of each input feature is also evaluated using an extension of integrated gradient across multiple dimensions.

**Limitations And Societal Impact:**

Especially for sensitive data, such as in the medical domain, the corpus examples need to be carefully anonymized.

**Main Review:**

While explaining neural networks through a combination of examples was introduced earlier, the ability to use a corpus that differs from the training data is, to the best of my knowledge, novel (unless I missed some important work). The integrated Jacobian is a fairly straightforward extension of integrated gradients to multiple dimensions.

It is great that the approach can be employed with a variety of corpora for explainability. However, the authors should also have evaluated the quality of corpus decomposition (section 3.1) when the training data is used.

The experiments cover example decomposition reasonably well, but pay relatively little attention to the feature explanations (using the integrated Jacobian). The authors show the contribution of the features in a few figures, but there is very limited discussion.

In addition to techniques used for black-box model explainability, I would suggest that the authors also describe approaches to directly make models more interpretable, such as glass-box models.

The paper is generally fairly clear, although there are numerous typos: L5 centred, L44 paterns, L96 example (should be plural), L97 accesible, L98 exaplanations, and likely others.

The paper has the potential to be significant as explainability can be an important consideration for the release of machine learning products. The methods are also fairly simple, which could accelerate their adoption by machine learning researchers and practitioners.

[Post author response] After reading the other reviews and the author response, I have increased the score I assigned to this submission.

My remark about using the training data was unclear. I meant using the entire training data as the corpus for SimplEx decomposition. This point was partially addressed in the response to reviewers QnMz and coXq with corpora up to 1000 examples.

The additional experiment in the author response helps justify the use of the integrated Jacobians over integrated gradients (although an even more thorough exploration would be preferable).

I am also satisfied by the authors' commitment to discuss some related work in more detail.

Another typo I saw when reading the paper again: L249 "bellow"

**Time Spent Reviewing:**

4

---

> ### Author Response · Authors · 2021-08-09
> **Response to Reviewer Zgwu**
>
> Thank you for your thoughtful comments and suggestions. We give answers to each in turn.
>
> ---
>
> ### (1) Reproducing the experiments with training data
>
> This is indeed a good remark. In our paper, we sample the corpus $\mathcal{C}$ from the training set to explain test examples from a separate test set. This simulates a typical use of SimplEx where we explain the model predictions and representations for examples unseen by the model. However, a similar experiment can perfectly be performed if we sample both the corpus $\mathcal{C}$ and the test examples from the training set. We have reproduced the Precision experiment (described in Section 3.1 of the main paper and Section 2.1 of the supplementary material) in this new setting. The results for MNIST are reported in the below Table (mean +/- standard deviation across 5 runs):
>
> |  Method |  $R^2_{\mathcal{H}} (K=2)$ 	|  $R^2_{\mathcal{H}} (K=5)$ | $R^2_{\mathcal{H}} (K=10)$ |     $R^2_{\mathcal{H}} (K=50)$ |
> | --- |	--- | --- | --- | --- |
> |   	SimplEx |   0.94 +/- 0.00 | 0.97 +/- 0.00 | 0.98 +/- 0.00 | 0.98 +/- 0.00 | 0.98 +/- 0.00 |
> |   	KNN Uniform|   0.94 +/- 0.00 | 0.94 +/- 0.01 | 0.94 +/- 0.01 |  0.90 +/- 0.01 |
> |   	KNN Distance |   0.94 +/- 0.00 | 0.95 +/- 0.01 | 0.95 +/- 0.01 | 0.92 +/- 0.01 |
>
>   We do the same for the prostate cancer dataset in the below Table (mean +/- standard deviation across 5 runs):
>
>
> |  Method |  $R^2_{\mathcal{H}} (K=2)$ 	|  $R^2_{\mathcal{H}} (K=5)$ | $R^2_{\mathcal{H}} (K=10)$ |     $R^2_{\mathcal{H}} (K=50)$ |
> |  ---   | ---   | --- | --- | --- |
> |   	SimplEx |   0.92 +/- 0.02 | 0.95 +/- 0.02 | 0.97 +/- 0.02 | 0.97 +/- 0.02 |
> |   	KNN Uniform|   0.92 +/- 0.02 | 0.90 +/- 0.02 | 0.90 +/- 0.02 |  0.67 +/- 0.08 |
> |   	KNN Distance |   0.92 +/- 0.02 | 0.91 +/- 0.02 | 0.89 +/- 0.02 | 0.82 +/- 0.08 |
>
>
> The results are consistent with what we obtained by using a separate test set in Section 3.1 of the main paper and thus the same conclusions hold.
>
> ---
>
> ### (2) Evaluation of Integrated Jacobians
>
> Thank you for pointing this out, we agree that additional discussions about Integrated Jacobians would be beneficial for the paper. Please allow us to first draw a subtle but important distinction between SimplEx and traditional saliency methods. Using the same notations as in the paper, suppose we want to explain the prediction $\textbf{f}(\textbf{x})$ for an input $\textbf{x} \in \mathcal{X} \subset \mathbb{R}^{d_X}$.
>
> Typical feature saliency methods (like Integrated Gradient) will attribute an importance score for each feature. These scores can be put together in an attribution vector $\textbf{a} \in \mathbb{R}^{d_X}$ of the same shape as $\textbf{x}$, where $a_i$ represents the importance of feature $x_i$ for the black-box $\textbf{f}$ to issue its prediction for each $i \in [d_X]$. What is crucial to notice here is that each saliency score is relative to the features of the input $\textbf{x}$ *only*.
>
> The saliency scores that SimplEx attributes at each feature carry a different meaning. We want to reconstruct the prediction $\textbf{f}(\textbf{x})$ in terms of a corpus $\mathcal{C} = \\{ \textbf{x}^c \mid c\in [C] \\}$. In this way, we attribute an importance score $p^c_i \in \mathbb{R}$ (the Jacobian Projection) to each feature $x^c_i$ of each corpus example $\textbf{x}^c$ for each $c\in [C]$. This score reflects how important is each of this feature in reconstructing the latent representation (and hence the prediction) of $\textbf{x}$ by using the examples for the corpus $\mathcal{C}$. In this way, each saliency score is relative to the features of *both* the $\textbf{x}$ and a given corpus example $\textbf{x}^c$.
>
> From this discussion, we see that the feature importance scores of traditional saliency methods describe something quite different from what SimplEx provides.  To check if SimplEx's Jacobian Projections are indeed a better measure of the importance of some features in constructing a latent representation, we propose the following procedure:
>
> 1. In the MNIST setting described in Section 3.1 of the main paper, we start with a corpus $\mathcal{C}$  of size  $C = 500$. We build a corpus approximation for an example $\textbf{x} \in \mathcal{X}$ with latent representation $\textbf{h} = \textbf{g}(\textbf{x}) \in \mathcal{H}$. The precision of this approximation is reflected by its corpus residual $r_{\mathcal{C}} (\textbf{h})$.
> 2. For each corpus example $\textbf{x}^c \in \mathcal{C}$, we would like to identify the features that are the most important in constructing the corpus decomposition of $\textbf{h}$. With SimplEx, this is reflected by the Jacobian Projections $p^c_i$. We evaluate these scores for each feature $i \in [d_X]$ of each corpus example  $c \in [C]$.
> 3. As a baseline for our experiment, we use Integrated Gradient, which is close in spirit to our method. In a similar fashion, we compute the Integrated Gradients $IG^c_i$ for each feature $i \in [d_X]$ of each corpus example  $c \in [C]$.
> 4. For each corpus image $\textbf{x}^c \in \mathcal{C}$, we select the $n$ most important pixels according to the Jacobian Projections and Integrated Gradients. In each case, we build a mask $\textbf{m}^c$ that replaces these $n$ most important pixels by black pixels. This yields a corrupted corpus image $\textbf{x}^c_{cor} = \textbf{m}^c \odot \textbf{x}^c$, where $\odot$ denotes the pointwise multiplication. By corrupting all the corpus images according to the Jacobian Projections and Integrated Gradients, we obtain two corrupted corpora $\mathcal{C}^{JP}$ and $\mathcal{C}^{IG}$, where JP and IG are abbreviations for Jacobian Projection and Integrated Gradient respectively.
> 5. We reproduce the first step with the two corrupted corpora, this yields residuals $r_{\mathcal{C}^{JP}}(\textbf{h})$ and $r_{\mathcal{C}^{IG}}(\textbf{h})$. We then compare the effectiveness of the corpus corruption by measuring the increase in the corpus residual caused by the corruption. This is reflected by the metric $\delta^{JP} = r_{\mathcal{C}^{JP}}(\textbf{h}) - r_{\mathcal{C}}(\textbf{h})$ and similarity for Integrated Gradient. A higher value for this metric indicates that the features selected by the saliency method are more salient for the corpus to produce a good approximation of $\textbf{h}$ in latent space.
>
> We repeat this experiment for 100 test examples and for different numbers  $n$ of masked pixels. The results are reported in the below table. We report the first quartile (Q1), the median (Q2) and the third quartile (Q3) for each metric.
>
>
> | $n$  | 1 | 5 |  10| 50 |
> |---|---|---|---|---|
> | $\delta^{JP}:Q_1$ | 0.5 | 3.5  | 5.9 | 10.6 |
> | $\delta^{IG}:Q_1$ | 0.0 | 1.5 | 4.0 | 9.8 |
> | $\delta^{JP}:Q_2$ | 1  | 4.9  | 7.6 | 12.8 |
> | $\delta^{IG}:Q_2$ | 0.3 | 3.0 | 5.8 | 12.0 |
> | $\delta^{JP}:Q_3$ | 1.9 | 6.6 | 9.7 | 15.2 |
> | $\delta^{IG}:Q_3$ | 0.9 | 4.5 | 7.4 | 13.9 |
>
>
> We observe that the corruptions induced by the Jacobian Projections are significantly more impactful. This is consistent with our above discussion and demonstrates that Jacobian Projections are more suitable to measure the importance of features when performing a latent space reconstruction.
>
> ---
>
> ### (3) Discussion on glass-box models
>
> While our main focus is on post-hoc explainability, we agree that a discussion on glass-box models would be a good addition to the revised manuscript. In particular, it would be interesting to discuss the rationale underlying each approach. For instance, glass-box models typically offer inherent interpretability by restricting to a model class with a more limited approximation power. On the other hand, post-hoc methods such as SimplEx aim at providing explanations without sacrificing the model complexity. There are very good discussions on this subject in the literature (see [1,2] for instance). We will make sure to add a discussion on this basis in the revised manuscript.
>
> [1] Arrieta, A. et al. “Explainable Artificial Intelligence (XAI): Concepts, Taxonomies, Opportunities and Challenges   		toward Responsible AI.” _ArXiv_ abs/1910.10045 (2020): n. pag.
>
> [2] Rai, A. Explainable AI: from black box to glass box. _J. of the Acad. Mark. Sci._  **48,** 137–141 (2020).
>
> ---
>
> ### (4) Suggested revisions
> Many thanks for the suggestions, we will correct the typos in the revised version.

---

### Official Review · Reviewer_h3ev · 2021-07-18

**Rating:** 8
**Confidence:** 4

**Summary:**

The main contribution of this paper is SimplEx: a method that aims to explain a test prediction by decomposing it into the weighted sum of nearest neighbors (from a user-specified corpus) in the latent space. They claim this method is more precise and robust than existing explanation-by-example-comparison methods (namely, KNN-like explanations) and distinguish this method from (often gradient-based) feature saliency methods. The paper builds off work that try to explain model predictions that use comparisons in latent variable space (e.g., Concept Activation Vectors, Deep KNN). SimplEx works PROVIDED the model has a hidden layer that linearly maps to the output (this is the layer that is used for explaining predictions).

- This constraint is significant for accurate weighting of examples. Note that if this is not satisfied by the model (say for instance a softmax is added right before the final output), it is possible to consider an earlier layer that satisfies the linear-mapping constraint as the output

Note too that if the latent space spanned by the user uploaded corpus does not contain the latent representation of the test prediction of interest (it likely will not), the test's latent is projected into the space. This means that it is possible to use this method on any input, regardless of the corpus. We can then measure how good this approximation is by calculating the residual between the corpus span and the test latent.

We can then "interpret the shift in latent space as resulting from a shift in the input space." This is the crux selling point of the paper, and it is accomplished using integrated (and projected) jacobians. This technique also lets us see the contribution of a feature from a corpus example.

The authors then proceed to (briefly) characterize their model on MNIST against dKNN inspired methods, and show a use case on a clinical risk dataset.

They claim the following novelty:

1. Freedom to choose comparison corpus, no need for this to equal the training set
2. The instance-level decompositions are valid in latent and output space, and are more robust than other explainability methods

**Limitations And Societal Impact:**

The method presented in the paper is described well, and the included examples (particularly of patient classification) show that this method can be appealing to those users of AI models that want to understand why a model makes certain predictions. Their claim that this method can generalize across models and datasets gives it even more potential impact. This reviewer is particularly intrigued by their proposition for future work — using this method to understand latent representations in unsupervised learning.

However, the authors themselves did not conduct experiments to show all their claims. They said it can work on regression problems, but showed no examples of it. There are no experiments comparing the results of SimplEx decompositions to feature saliency methods. There are no user studies showing that this would be helpful in the real world. And, while reporting performance metrics on interpretability methods is not the standard, I believe it would be good to discuss this point.

I believe this paper deserves to be included in the NeurIPS proceedings, but its story could be improved by addressing the above.

**Main Review:**

The main contribution of this paper is SimplEx: a method that aims to explain a test prediction by decomposing it into the weighted sum of nearest neighbors (from a user-specified corpus) in the latent space. They claim this method is more precise and robust than existing explanation-by-example-comparison methods (namely, KNN-like explanations) and distinguish this method from (often gradient-based) feature saliency methods. The paper builds off work that try to explain model predictions that use comparisons in latent variable space (e.g., Concept Activation Vectors, Deep KNN). SimplEx works PROVIDED the model has a hidden layer that linearly maps to the output (this is the layer that is used for explaining predictions).

- This constraint is significant for accurate weighting of examples. Note that if this is not satisfied by the model (say for instance a softmax is added right before the final output), it is possible to consider an earlier layer that satisfies the linear-mapping constraint as the output

Note too that if the latent space spanned by the user uploaded corpus does not contain the latent representation of the test prediction of interest (it likely will not), the test's latent is projected into the space. This means that it is possible to use this method on any input, regardless of the corpus. We can then measure how good this approximation is by calculating the residual between the corpus span and the test latent.

We can then "interpret the shift in latent space as resulting from a shift in the input space." This is the crux selling point of the paper, and it is accomplished using integrated (and projected) jacobians. This technique also lets us see the contribution of a feature from a corpus example.

The authors then proceed to (briefly) characterize their model on MNIST against dKNN inspired methods, and show a use case on a clinical risk dataset.

They claim the following novelty:

1. Freedom to choose comparison corpus, no need for this to equal the training set
2. The decompositions are valid in latent and output space

# Main Review

Provide a full review of the submission, including its originality, quality, clarity, and significance. See [https://neurips.cc/Conferences/2021/Reviewer-Guidelines](https://neurips.cc/Conferences/2021/Reviewer-Guidelines) for guidance on questions to address in your review, and /faq for how to incorporate Markdown and LaTeX into your review.

First, a brief note on Claim 1 — letting users choose the corpus of examples to decompose model predictions into is not particularly novel. The mentioned dKNN method could be easily used with any user-specified corpus, as the authors do in their "Experiments" section.

The submission is well written, and it dives deeply into the problem of understanding model prediction as a combination of human-interpretable examples AND features. This reviewer is not aware of other literature that comprehensively bridges these two methods of explainability, and I believe many fields in AI Interpretability can utilize or be inspired by this effort.

As it stands, the authors have this reviewer convinced that this method is useful in explaining many simple Classification problems.

- Authors show that this method is better than using KNN to explain, since we don't want to include irrelevant corpus examples.
- Authors show that it can be used to inform users if the prediction is likely correct or not by relating it to examples in the corpus that were either correct or not.
- Authors show this working on both tabular data and image data (although interpreting image features is a bit more challenging)

In its current state, the submission lacks:

1. Accompanying code to play with results (authors have reported it will be published eventually)
2. Stories showing this method's effectiveness on problems beyond simple classification and architectures beyond established MLP and CNNs.
3. Quantitative (and even qualitative) comparisons to existing methods. For example, showing this method working on MNIST is actually rather boring, since no comparable results from other saliency methods are shown.
    - Additionally, this could take the form of a quick user study involving real doctors/nurses on the provided clinical risk dataset.

I also imagine this method is significantly slower than comparable ones (like dKNN) since it requires calculation of the integrated jacobian for every test example. This would then require larger compute to explain a large batch of predictions at once. It would be good if the authors included some performance metrics in their paper.

### Additional Comments

1. The introductory paragraph uses large, sweeping sentences that are both unprofessional in an academic paper and distracting from the point. Please reduce the wordiness.
2. Please check the paper for grammatical English errors. There are many instances where plural nouns are incorrectly substituted by their singular form, and several sentences early in the paper that I had trouble understanding:
    - "paterns", "exaplanations", "accesible", "developped"
    - [L188] "we do no longer need"
    - [L213] "an a corpus example"
3. Figures with screenshots of tabular data are difficult to read without zooming in.
4. I STRONGLY recommend choosing a color scheme other than green/red for Figure [3,7,8] for accessibility reasons. Consider a secondary encoding, or solutions proposed [here](https://www.visualisingdata.com/2019/08/five-ways-to-design-for-red-green-colour-blindness/#:~:text=The%20pink%2Dred%20through%20to,green%20hues%20used%20by%20default.) or [here](https://www.visualisingdata.com/2015/11/colour-swatch-alternatives-to-green-and-red/)

**Time Spent Reviewing:**

10

---

> ### Author Response · Authors · 2021-08-09
> **Response to Reviewer h3ev [Part 2/2]**
>
> ### (4) Speed in comparison with dKNN
>
> We believe that a fair comparison between SimplEx and dKNN in terms of speed should exclude the evaluation of the Integrated Jacobians. Indeed, SimplEx uses this step to attribute an importance score to each feature of each example from the corpus. This functionality is absent from the pure dKNN approach that only involves finding the K-Nearest corpus neighbours in latent space. Let us now give a more concrete discussion of the comparable method. Once again, we want to explain the black-box prediction $\textbf{f}(\textbf{x})$ for an input $\textbf{x} \in \mathcal{X} \subset \mathbb{R}^{d_X}$.
>
> A dKNN approach involves $C$ calls of the black-box model (we need to evaluate the latent representation $\textbf{h}^c$ of each corpus example $\textbf{x}^c$ for $c \in [C]$). Then, the dKNN ranks the corpus examples by decreasing distance with respect to the latent representation $\textbf{h}$ of $\textbf{x}$ (i.e. decreasing $\vert\vert \textbf{h} - \textbf{h}^c \vert\vert$) to select the K-Nearest Neighbours. Then, the weights $w^c$ are deduced without any additional computation.
>
> The part of SimplEx that performs a similar analysis (i.e. computing the weights) corresponds to Algorithm 1 of the Supplementary Material. Concretely, this method also requires $C$ calls of the black-box to evaluate the latent representation $\textbf{h}^c$ for each $c \in [C]$. Once this is done, SimplEx finds the optimal weights $w^c$ without making any additional call to the black-box. Solving the optimization problem corresponding to Equation (1) from the main paper is more expensive than ranking the corpus example as is done for dKNN. However, it corresponds to solving a trivial convex optimization problem that does not require any further call to the black-box.
>
> On the below table, we have provided the time taken to evaluate the weights $w^c$ for one test sample and a corpus of size $C=1000$ with our MNIST classifier model described in Table 3 of the supplementary material:
>
>
> |   	Method |  Computation time per test sample (seconds)  	|
> |---				|	---    |
> |   	SimplEx (corpus decomposition)	 |   $(8.1  \pm 0.1 ) \cdot 10^{-3}$ 	|
> |  dKNN |  $(3.1  \pm 0.2 ) \cdot 10^{-5}$ 	|
>
> We observe that, indeed, learning the weights costs extra computation time. However, we believe that the gain in precision reported in Section 3.1 of the main paper and Section 2.4.1 of the Supplementary Material clearly justifies that extra cost. Moreover, we believe that both approaches are fast enough for most practical purposes.
>
>
> The evaluation of the Jacobian Projections is more expensive and took $(5.6  \pm 0.0 ) \cdot 10^{-1}$ seconds per sample in the previous setting. Even though this computation time is higher, it is still reasonable when compared to other methods such as influence functions. In the same setting, it took $1.8  \pm 0.0$ seconds per sample to compute the associated corpus influence functions. This stems from the fact that our method does not require the estimation of any Hessian/Hessian-vector product of the model with respect to its parameters.
>
>
> ---
>
> ### (5) Suggested revisions
> Many thanks for the suggestions, we will reduce the wordiness in the introduction, correct the typos and make the figures more  accessible in the revised version.
>
> ---

---

> > ### Comment · Reviewer_h3ev · 2021-08-24
> > **Final thoughts**
> >
> > Thank you for your clarifications. I found your responses to my original review improved my understanding of your method, and I strongly believe this technique will improve trust and interpretability in AI methods. I suggest that your analysis of the speed & efficiency of this method be included in a section of the supplementary material, but mention and link to this new section from the main paper. For researchers and engineers who build interpretability tools for different algorithms, it is important to quantify the speed tradeoffs between different methods running in live systems.

---

> > > ### Author Response · Authors · 2021-08-30
> > > **Final anwer**
> > >
> > > Many thanks for this feedback and for this thorough review.
> > > We will definitely include this section to the supplementary material and a link in the main paper.

---

> ### Author Response · Authors · 2021-08-09
> **Response to Reviewer h3ev [Part 1/2]**
>
> Thank you for your thoughtful comments and suggestions. We give answers to each in turn.
>
> ---
>
> ### (1) Effectiveness beyond simple classification
>
> We agree that the tasks presented in the experiment section of the paper are quite standard.  We believe that the experiments from Section 2.4 of the Supplementary Material show that our method yields good results for a time series forecasting problem. In this experiment, the task of interest is a regression problem (predicting the value of the time series at the next time step) and we use a different type of model architecture (LSTM).  We agree that this experiment is not sufficiently emphasized in the main paper, we will make sure to mention it more clearly in the revised manuscript.
>
> ---
> ### (2) Comparison with feature saliency methods
>
>
> Thank you for pointing this out, please allow us to first draw a subtle but important distinction between SimplEx and traditional saliency methods. Using the same notations as in the paper, suppose we want to explain the prediction $\textbf{f}(\textbf{x})$ for an input $\textbf{x} \in \mathcal{X} \subset \mathbb{R}^{d_X}$.
>
> Typical feature saliency methods (like Integrated Gradient) will attribute an importance score for each feature. These scores can be put together in an attribution vector $\textbf{a} \in \mathbb{R}^{d_X}$ of the same shape as $\textbf{x}$, where $a_i$ represents the importance of feature $x_i$ for the black-box $\textbf{f}$ to issue its prediction for each $i \in [d_X]$. What is crucial to notice here is that each saliency score is relative to the features of the input $\textbf{x}$ *only*.
>
> The saliency scores that SimplEx attributes at each feature carry a different meaning. We want to reconstruct the prediction $\textbf{f}(\textbf{x})$ in terms of a corpus $\mathcal{C} = \\{ \textbf{x}^c \mid c\in [C] \\}$. In this way, we attribute an importance score $p^c_i \in \mathbb{R}$ (the Jacobian Projection) to each feature $x^c_i$ of each corpus example $\textbf{x}^c$ for each $c\in [C]$. This score reflects how important is each of this feature in reconstructing the latent representation (and hence the prediction) of $\textbf{x}$ by using the examples for the corpus $\mathcal{C}$. In this way, each saliency score is relative to the features of *both* $\textbf{x}$ and a given corpus example $\textbf{x}^c$.
>
> From this discussion, we see that the feature importance scores of traditional saliency methods describe something quite different from what SimplEx provides.  To check if SimplEx's Jacobian Projections are indeed a better measure of the importance of some features in constructing a latent representation, we propose the following procedure:
>
> 1. In the MNIST setting described in Section 3.1 of the main paper, we start with a corpus $\mathcal{C}$ of size $C = 500$. We build a corpus approximation for an example $\textbf{x} \in \mathcal{X}$ with latent representation $\textbf{h} = \textbf{g}(\textbf{x}) \in \mathcal{H}$. The precision of this approximation is reflected by its corpus residual $r_{\mathcal{C}}(\textbf{h})$.
> 2. For each corpus example $\textbf{x}^c \in \mathcal{C}$, we would like to identify the features that are the most important in constructing the corpus decomposition of $\textbf{h}$. With SimplEx, this is reflected by the Jacobian Projections $p^c_i$. We evaluate these scores for each feature $i \in [d_X]$ of each corpus example  $c \in [C]$.
> 3. As a baseline for our experiment, we use Integrated Gradient, which is close in spirit to our method. In a similar fashion, we compute the Integrated Gradients $IG^c_i$ for each feature $i \in [d_X]$ of each corpus example  $c \in [C]$.
> 4. For each corpus image $\textbf{x}^c \in \mathcal{C}$, we select the $n$ most important pixels according to the Jacobian Projections and Integrated Gradients. In each case, we build a mask $\textbf{m}^c$ that replaces these $n$ most important pixels by black pixels. This yields a corrupted corpus image $\textbf{x}^c_{cor} = \textbf{m}^c \odot \textbf{x}^c$, where $\odot$ denotes the pointwise multiplication. By corrupting all the corpus images according to the Jacobian Projections and Integrated Gradients, we obtain two corrupted corpora $\mathcal{C}^{JP}$ and $\mathcal{C}^{IG}$, where JP and IG are abbreviations for Jacobian Projection and Integrated Gradient respectively.
> 5. We reproduce the first step with the two corrupted corpora, this yields residuals $r_{\mathcal{C}^{JP}}(\textbf{h})$ and $r_{\mathcal{C}^{IG}}(\textbf{h})$. We then compare the effectiveness of the corpus corruption by measuring the increase in the corpus residual caused by the corruption. This is reflected by the metric $\delta^{JP} = r_{\mathcal{C}^{JP}}(\textbf{h}) - r_{\mathcal{C}}(\textbf{h})$ and similarity for Integrated Gradient. A higher value for this metric indicates that the features selected by the saliency method are more salient for the corpus to produce a good approximation of $\textbf{h}$ in latent space.
>
> We repeat this experiment for 100 test examples and for different numbers  $n$ of masked pixels. The results are reported in the below table. We report the first quartile (Q1), the median (Q2) and the third quartile (Q3) for each metric.
>
>
> | $n$  | 1 | 5 |  10| 50 |
> |---|---|---|---|---|
> | $\delta^{JP}:Q_1$ | 0.5 | 3.5  | 5.9 | 10.6 |
> | $\delta^{IG}:Q_1$ | 0.0 | 1.5 | 4.0 | 9.8 |
> | $\delta^{JP}:Q_2$ | 1  | 4.9  | 7.6 | 12.8 |
> | $\delta^{IG}:Q_2$ | 0.3 | 3.0 | 5.8 | 12.0 |
> | $\delta^{JP}:Q_3$ | 1.9 | 6.6 | 9.7 | 15.2 |
> | $\delta^{IG}:Q_3$ | 0.9 | 4.5 | 7.4 | 13.9 |
>
>
> We observe that the corruptions induced by the Jacobian Projections are significantly more impactful. This is consistent with our above discussion and demonstrates that Jacobian Projections are more suitable to measure the importance of features when performing a latent space reconstruction.
>
> ---
>
> ### (3) User study
>
> As suggested, we have conducted a small scale user study with SimplEx. The purpose of this study was to identify if the functionalities introduced by SimplEx are interesting for the clinicians. In total, 10 clinicians took part in the study.
>
> Let us now describe the study. With the SEER Prostate Cancer dataset that is described in the paper, we have performed a SimplEx corpus decomposition such as the one presented in Figure 1 of the main paper. The decomposition involved 1 test patient that we called Joe and 2 corpus patients that we called Bill and Max. Our classification model predicted that Joe will die of his prostate cancer. Bill died of his prostate cancer and Max survived. The SimplEx corpus weights were as follows: 66% for Bill, 34% for Max. For both Bill and Max, the Jacobian Projections were given and presented as a measure of importance for each of their features in order to relate them to Joe.
>
> After presenting this explanation to the clinician, we gradually brought their attention to its various components. We made several statements related to SimplEx's functionalities and asked the clinicians if they agree/disagree on a scale from 0 to 5, where 0 corresponds to strongly disagreeing, 3 corresponds to a neutral opinion and 5 corresponds to strongly agreeing.
>
> The first two statements were related to the weights appearing in the corpus decomposition. The purpose was to determine if those are important for the clinicians and if there is an additional value in learning these weights, as is done in SimplEx. The first statement was the following: “The value of the weights in the corpus decomposition is important”. The results were the following: 6 of the clinicians agreed (1 strongly), 1 remained neutral and 3 disagreed (1 strongly). The second statement was the following: “Some valuable information is lost in setting the weights to a uniform value” (the doctors are given the KNN Uniform equivalent of SimplEx's explanation in the presented case). The results were the following:  5 of the clinicians agreed (3 strongly), 3 remained neutral, 2 strongly disagreed. We conclude that the majority of the clinicians found the weights to be important. Most of them found that hard-coding the weights as in the KNN Uniform baseline hides some valuable information.
>
> The third statement was related to  the Jacobian Projections. The purpose was to determine if the Jacobian Projections provide valuable information for interpretability. The statement was the following: “Knowing which feature increases the similarity/discrepancy between two patients is important”. The results were the following: 9 of the clinicians agreed (5 strongly), 1 disagreed. We conclude that the Jacobian Projections constitute a crucial part of SimplEx's explanations.
>
> The fourth statement was related to the freedom of choosing the corpus. The purpose was to determine if the flexibility of SimplEx is useful in practice. The  statement was the following: “It is important for the clinician to be able to choose the patients in the corpus that is used for comparison”. The results were the following: 4 of the clinicians agreed (1 strongly), 1 remained neutral, 5 disagreed (3 strongly). Clearly, the clinicians are more divided on this point. However, this additional freedom offered by SimplEx comes at no cost. A clinician that desires explanations in terms of patients they are familiar with can use their own copus. A clinician that is happy with explanations in terms of any patients can use a corpus sampled from training data.
>
> The last statement was related to the use of SimplEx in order to anticipate misclassification, as it is suggested in Section 3.2 of the main paper. The  statement was the following:  “If Bill had not died due to his prostate cancer, this would cast doubt on the mortality predicted for Joe”. The results were the following: 6 of the clinicians agreed (2 strongly), 1 remained neutral, 3 disagreed (2 strongly). We conclude that, for the majority of the clinicians, SimplEx's explanations affect their confidence in the model's prediction.
>
>
> ---

---

### Decision · Program_Chairs · 2021-09-27

**Decision:**

Accept (Spotlight)

**Comment:**

This paper proposes a method for explaining machine learning models by decomposing the continuous representation of a test example into a combination of examples from a given prototype corpus.

Reviewers find the method convincing, powerful (bridging two large framework of explainable ML: feature-based and example-based), well presented and well demonstrated. The authors have thoroughly addressed all outstanding concerns in the discussion period, including a small user study. I strongly urge the authors to take the time to implement the corresponding improvements in the manuscript.